# Parabolic Approximation Line Search for DNNs

**Maximus Mutschler and Andreas Zell**
University of Tübingen
Sand 1, D-72076 Tübingen, Germany
{maximus.mutschler, andreas.zell}@uni-tuebingen.de

## Abstract

A major challenge in current optimization research for deep learning is to automatically find optimal step sizes for each update step. The optimal step size is closely related to the shape of the loss in the update step direction. However, this shape has not yet been examined in detail. This work shows empirically that the batch loss over lines in negative gradient direction is mostly convex locally and well suited for one-dimensional parabolic approximations. By exploiting this parabolic property we introduce a simple and robust line search approach, which performs loss-shape dependent update steps. Our approach combines well-known methods such as parabolic approximation, line search and conjugate gradient, to perform efficiently. It surpasses other step size estimating methods and competes with common optimization methods on a large variety of experiments without the need of hand-designed step size schedules. Thus, it is of interest for objectives where step-size schedules are unknown or do not perform well. Our extensive evaluation includes multiple comprehensive hyperparameter grid searches on several datasets and architectures. Finally, we provide a general investigation of exact line searches in the context of batch losses and exact losses, including their relation to our line search approach.

## 1 Introduction

Automatic determination of optimal step sizes for each update step of stochastic gradient descent is a major challenge in current optimization research for deep learning [3, 5, 12, 29, 38, 43, 46, 50, 58]. One default approach to tackle this challenge is to apply line search methods. Several of these have been introduced for Deep Learning [12, 29, 38, 43, 58]. However, these approaches have not analyzed the shape of the loss functions in update step direction in detail, which is important, since the optimal step size stands in strong relation to this shape. To shed light on this, our work empirically analyses the shape of the loss function in update step direction for deep learning scenarios often considered in optimization. We further elaborate the properties found to define a simple, competitive, empirically justified optimizer.

Our contributions are as follows: **1:** Empirical analysis suggests that the loss function in negative gradient direction mostly shows locally convex shapes. Furthermore, we show that parabolic approximations are well suited to estimate the minima in these directions (Section 3). **2:** Exploiting the parabolic property, we build a simple line search optimizer which constructs its own loss function dependent learning rate schedule. The performance of our optimization method is extensively analyzed, including a comprehensive comparison to other optimization methods (Sections 4,5). **3:** We provide a convergence analysis which backs our empirical results, under strong assumptions (Section 4.4). **4:** We provide a general investigation of exact line searches on batch losses and their relation to line searches on the exact loss as well as their relation to our line search approach (Section 6) and, finally, analyze the relation of our approach to interpolation (Section 7).

The empirical loss $\mathcal{L}$ is defined as the average over realizations of a batch-wise loss function $L$: $\mathcal{L}(\theta) : \mathbb{R}^m \to \mathbb{R}, \theta \mapsto n^{-1} \sum_{i=1}^{n} L(\mathbf{x}_i; \theta)$ with $n$ being the amount of batches, $\mathbf{x}_i$ denotes a batch of a dataset and $\theta \in \mathbb{R}^m$ denotes the parameters to be optimized. Note, that we consider a sample as one batch of multiple inputs. We denote $L(\mathbf{x}_t; \theta_t)$ the batch loss of a batch $\mathbf{x}$ at optimization step $t$. In this work, we consider $L(\mathbf{x}_t; \theta_t)$ in negative gradient direction:

$$l_t(s) : \mathbb{R} \to \mathbb{R}, s \mapsto L(\mathbf{x}_t; \theta_t + s \cdot \frac{-\mathbf{g}_t}{||\mathbf{g}_t||}) \tag{1}$$

where $\mathbf{g}_t$ is $\nabla_{\theta_t} L(\mathbf{x}_t; \theta_t)$. For simplification, we denote $l_t(s)$ a line function or vertical cross section and $s$ a step on this line. The motivation of our work builds upon the following assumption:

**Assumption 1.** *(Informal) The position $\theta_{min} = \theta_t + s_{min} \frac{-\mathbf{g}_t}{||\mathbf{g}_t||}$ of a minimum of $l_t$ is a well enough estimator for the position of the minimum of the empirical loss $\mathcal{L}$ on the same line to perform a successful optimization process.*

We empirically analyze Assumption 1 further in section 6.

## 2 Related work

Our optimization approach is based on well-known methods, such as line search, the non linear conjugate gradient method and quadratic approximation, which can be found in Numerical Optimization [28], which, in addition, describes a similar line search routine for the deterministic setting. The concept of parabolic approximations is also exploited by the well known line search of More and Thunte [40]. Our work contrasts common optimization approaches in deep learning by directly exploiting the parabolic property (see Section 3) of vertical cross sections of the batch loss. Similarly, *SGD-HD* [3] performs update steps towards the minimum on vertical cross sections of the batch loss, by performing gradient descent on the learning rate. Concurrently, [10] explored a similar direction as this work by analyzing possible line search approximations for DNN loss landscapes, but does not exploit these for optimization.

The recently published *Stochastic Line-Search* (*SLS*) [58] is an optimized backtracking line search based on the Armijo condition, which samples, like our approach, additional batch losses from the same batch and checks the Armijo condition on these. [58] assumes that the model interpolates the data. Formally, this implies that the gradient at a minimum of the empirical loss is 0 for the empirical loss as well as for all batch (sample) losses. [12] also uses a backtracking Armijo line search, but with the aim to regulate the optimal batch size. *SLS* exhibits competitive performance against multiple optimizers on several DNN tasks. [43] introduces a related idea but does not provide empirical results for DNNs.

The methodically appealing but complex *Probabilistic Line Search* (*PLS*) [38] and *Gradient Only Line Search* (*GOLS1*) [29] are considering a discontinuous stochastic loss function. *GOLS1* searches for a minimum on lines by searching for a sign change of the first directional derivative in search direction. *PLS* optimizes on lines of a stochastic loss function by approximating it with a Gaussian Process surrogate and exploiting a probabilistic formulation of the Wolf conditions. Both approaches show that they can optimize successfully on several machine learning problems and can compete against plain *SGD*.

From the perspective of assumptions about the shape of the loss landscape, second order methods such as *oLBFGS* [53], *KFRA* [7], *L-SR1* [45], *QUICKPROP* [15], *S-LSR1* [4], and *KFAC* [39] generally assume that the loss function can be approximated locally by a parabola of the same dimension as the loss function. Adaptive methods such as *SGD* with momentum [49], *ADAM* [30], *ADAGRAD* [14], *ADABOUND* [37], *AMSGRAD* [47] or *RMSProp* [57] focus more on the handling of noise than on shape assumptions. In addition, methods exist that approximate the loss function in specific directions: The *L4* adaptation scheme [50] as well as *ALIG* [5] estimate step sizes by approximating the loss function linearly in negative gradient direction, whereas our approach approximates the loss function parabolically in negative gradient direction.

Finally, *COCOB* [42] has to be mentioned, an alternative learning rate free approach, which automatically estimates step directions and sizes with a reward based coin betting concept.

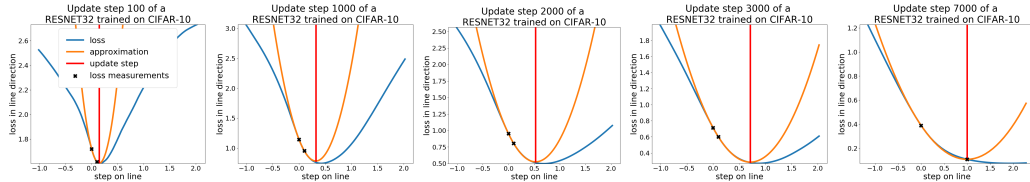

Figure 1: Representative batch losses on cross sections in negative normalized gradient direction (blue), parabolic approximations (orange) and the position of the approximated minima (red). Further plots are provided in Appendix A.

# 3 Empirical analysis of the shape of batch losses on vertical cross sections

In this section we analyze line functions (see Eq. 1) during the training of multiple architectures and show that they locally exhibit mostly convex shapes, which are well suited for parabolic approximations. We focus on CIFAR-10, as it is extensively analyzed in optimization research for deep learning. However, on random samples of MNIST, CIFAR-100 and ImageNet we observed the same results. We analyzed cross sections of 4 common used architectures in detail. To do so, we evaluated the cross sections of the first 10000 update steps for each architecture. For each cross section we sampled 50 losses and performed a parabolic approximation (see Section 4). An unbiased selection of our results on a ResNet32 is shown in Figure 1. Further results are given in Appendix A. In accordance with [59], we conclude that the analyzed cross sections tend to be locally convex. In addition, one-dimensional parabolic approximations of the form $f(s) = as^2 + bs + c$ with $a \neq 0$ are well suited to estimate the position of a minimum on such cross sections. To substantiate the later observation, we analyzed the angle between the line direction and the gradient at the estimated minimum during training. A position is a local extremum or saddle point of the cross section if and only if the angle between the line direction and the gradient at the position is $90°$, if measured on the same batch. [1] As shown in Figures 2 and 3, this property holds well for several architectures trained on MNIST, CIFAR-10, CIFAR-100 and ImageNet. The property fits best for MNIST and gets worse for more complex tasks such as ImageNet. We have to note, that measuring step sizes and update step adaptations factors (see Sections 4.1 and4.3) were chosen to fit the line functions decently. We can ensure that the extrema found are minima, since we additionally plotted the line function for each update step.

In addition, we analyzed vertical cross sections in conjugate like directions and random directions. Vertical cross section in conjugate like directions also tend to have convex shapes (see Appendix D.4 Figure 17 ). However, vertical cross sections in random directions rarely exhibit convex shapes.

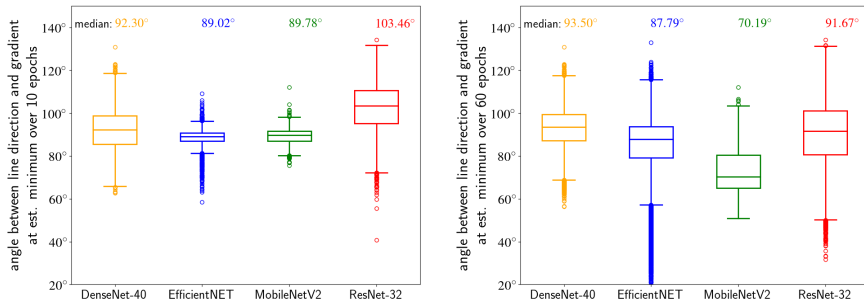

Figure 2: Angles between the line direction and the gradient at the estimated minimum measured on the same batch. If the angle is $90°$, the estimated minimum is a real local minimum. We know from additional line plots that the found extrema or saddle points are minima. Left: measurement over the first 10 epochs. Right: measurement over the first 60 epochs. Update step adaptation (see Section 4.3) is applied.

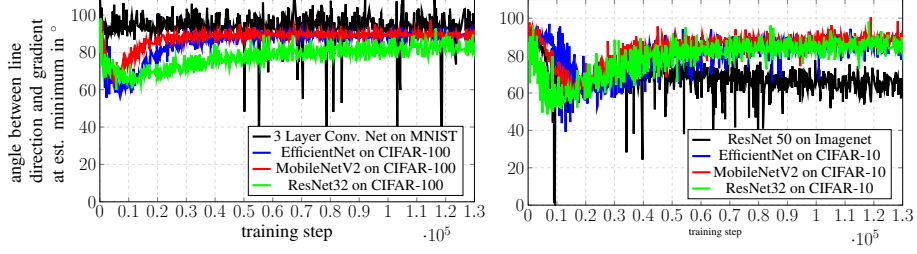

Figure 3: Angles between the line direction and the gradient at the estimated minimum measured on the same batch plotted over a whole training process on several networks and datasets. This figure clarifies that the parabolic property is also valid on further datasets and during the training process. It fits best for MNIST and becomes worse for ImageNet. Measuring step sizes and update step adaptations factors (see Sections 4.1,4.3) were used to fit the cross sections decently.

# 4 The line search algorithm

By exploiting the property, that parabolic approximations are well suited to estimate the position of minima on line functions, we introduce **P**arabolic **A**pproximation **L**ine Search (*PAL*). This simple approach combines well-known methods from basic optimization such as parabolic approximation and line search [28], to perform an efficient line search. We note, that the general idea of this method can be applied to any optimizer that provides an update step direction.

## 4.1 Parameter update rule

An intuitive explanation of *PAL*'s parameter update rule based on a parabolic approximation is given in Figure 4. Since $l_t(s)$ (see Eq.1) is assumed to exhibit a convex and almost parabolic shape, we approximate it with $\hat{l}_t(s) = as^2 + bs + c$ with $a \neq 0$ and $a, b, c \in \mathbb{R}$. Consequently, we need three measurements to define $a, b$ and $c$. Those are given by the current loss $l_t(0)$, the derivative in gradient direction $l'_t(0) = -||\mathbf{g}_t||$ (see Eq. 4) and an additional loss $l_t(\mu)$ with measuring distance $\mu \in \mathbb{R}^+$. We get $a = \frac{l_t(\mu) - l_t(0) - l'_t(0)\mu}{\mu^2}$, $b = l'_t(0)$, and $c = l_t(0)$. The update step $s_{upd}$ to the minimum of the parabolic approximation $\hat{l}_t(s)$ is thus given by:

$$s_{upd_t} = -\frac{\hat{l}'_t(0)}{\hat{l}''_t(0)} = -\frac{b}{2a} = \frac{-l'_t(0)}{2\frac{l_t(\mu) - l_t(0) - l'_t(0)\mu}{\mu^2}} \quad (2)$$

Note, that $\hat{l}''_t(0)$ is the second derivative of the approximated parabola and is only identical to the exact directional derivative $\frac{-\mathbf{g}_t}{||\mathbf{g}_t||} H(L(\mathbf{x}_t; \theta_t)) \frac{-\mathbf{g}_t^T}{||\mathbf{g}_t||}$ if the parabolic approximation fits. The normalization of the gradient to unit length (Eq.1) was chosen to have the measuring distance $\mu$ independent of the gradient size and of weight scaling. Note that two network inferences are required to determine $l_t(0)$ and $l_t(\mu)$. Consequently, *PAL* needs two forward passes and one backward pass through a model. Further on, the batch loss $L(\mathbf{x}_t; \theta_t)$ may include random components, but, to ensure con-

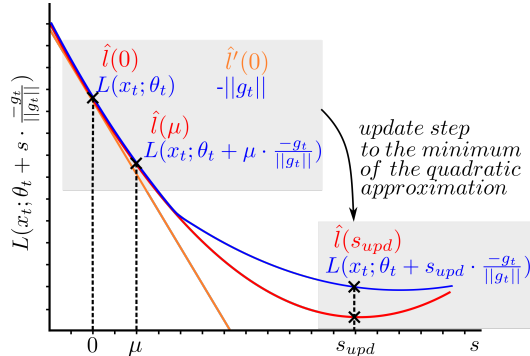

Figure 4: Basic idea of *PAL*'s parameter update rule. The blue curve is the cross section of the loss function in direction of the negative gradient at $L(\mathbf{x}_t; \theta_t)$. It is defined by $l(s) = L(\mathbf{x}_t; \theta_t + s \cdot \frac{-\mathbf{g}_t}{||\mathbf{g}_t||})$ where $g_t$ is $\nabla_{\theta_t} L(\mathbf{x}_t; \theta_t)$. The red curve is its parabolic approximation $\hat{l}(s)$. With $l(0), l(\mu)$ and $\mathbf{g}_t$ (orange), we have the three parameters needed to determine the update step $s_{upd}$ to the minimum of the parabolic approximation.

tinuity during one line search, drawn random numbers have to be reused for each value determination of $L$ at $t$ (e.g. for Dropout [55]). The memory required by *PAL* is similar to *SGD* with momentum, since only the last update direction has to be saved. A basic, well performing version of *PAL* is given in Algorithm 1.

---

**Algorithm 1** The basic version of our proposed line search algorithm. See Section 4 for details.

---

**Input:** $\mu$: measuring step size
**Input:** $L(x; \theta)$: loss function
**Input:** $\mathbf{x}$: list of input vectors
**Input:** $\theta_0$: initial parameter vector
1: $t \leftarrow 0$
2: **while** $\theta_t$ not converged **do**
3:     $l_0 \leftarrow L(\mathbf{x}_t; \theta_t)$     # $l_0 = l_t(0)$ see Eq. 1
4:     $g_t \leftarrow -\nabla_{\theta_t} L(\mathbf{x}_t; \theta_t)$
5:     $l_\mu \leftarrow L(\mathbf{x}_t; \theta_t + \mu \frac{g_t}{||g_t||})$
6:     $b \leftarrow -||g_t||$

7:     $a \leftarrow \frac{l_\mu - l_0 - b\mu}{\mu^2}$
8:     **if** proper curvature **then**
9:       $s_{upd} \leftarrow -\frac{b}{2a}$
10:    **else**
11:       # set $s_{upd}$ according to section 4.2
12:    **end if**
13:    $\theta_{t+1} \leftarrow \theta_t + s_{upd} \frac{g_t}{||g_t||}$
14:    $t \leftarrow t + 1$
15: **end while**
16: **return** $\theta_t$

---

## 4.2 Case discrimination of parabolic approximations

Since not all parabolic approximations are suitable for parameter update steps, the following cases are considered separately. Note that $b = l_t'(0)$ and $a = 0.5 l_t''(0)$. **1:** $a > 0$ and $b < 0$: parabolic approximation has a minimum in line direction, thus, the parameter update is done as described in Section 4.1. **2:** $a \leq 0$ and $b < 0$: parabolic approximation has a maximum in negative line direction, or is a line with negative slope. In those cases a parabolic approximation is inappropriate. $s_{upd}$ is set to $\mu$, since the second measured point has a lower loss than the first. **3:** Since $b = -||g_t||$ cannot be greater than 0, the only case left is an extremum at the current position ($l'(0) = 0$). In this case, no weight update is performed. However, the loss function is changed by the next batch. In accordance to Section 3, cases 2 and 3 appeared very rarely in our experiments.

## 4.3 Additions

We introduce multiple additions for Algorithm 1 to fine tune the performance and handle degenerate cases. We emphasize that our hyperparameter sensitivity analysis (Appendix D.6) suggests that the influence of the introduced hyperparameters on the optimizer's performance are low. Thus, they only need to be adapted to fine tune the results. The full version of *PAL* including all additions is given in Appendix B Algorithm 2.

**Direction adaptation:** Instead of following the direction of the negative gradient we follow an adapted conjugate-like direction $d_t$:

$$d_t = -\nabla_{\theta_t} L(\mathbf{x}_t; \theta_t) + \beta d_{t-1} \quad d_0 = -\nabla_{\theta_0} L(x_0; \theta_0) \tag{3}$$

with $\beta \in [0, 1]$. Since now an adapted direction is used, $l_t'(0)$ changes to:

$$l_t'(0) = \nabla_{\theta_t} L(\mathbf{x}_t; \theta_t) \frac{d_t}{||d_t||} \tag{4}$$

This approach aims to find a more optimal search direction than the negative gradient. We implemented and tested the formulas of Fletcher-Reeves [16], Polak-Ribière [48], Hestenes-Stiefel [24] and Dai-Yuan [11] to determine conjugate directions under the assumption that the loss function is a quadratic. However, choosing a constant $\beta$ of value 0.2 or 0.4 performs equally well. The influence of $\beta$ and dynamic update steps on *PAL*'s performance is discussed in Appendix D.5. In the analyzed scenario $\beta$ can both increase and decrease the performance, whereas, dynamic update steps mostly increase the performance. The combination of both is needed to achieve optimal results.

**Update step adaptation:** Our preliminary experiments revealed a systematic error caused by constantly approximating with slightly too narrow parabolas. Therefore, $s_{upd}$ is multiplied by a parameter $\alpha \geq 1$ (compare to Eq. 2). This is useful to estimate the position of the minimum on a line more exactly, but has minor effects on training performance.

**Maximum step size:** To hinder the algorithm from failing due to inaccurate parabolic approximations, we use a maximum step size $s_{max}$. The new update step is given by $min(s_{upd}, s_{max})$. However, most of our experiments with $s_{max} = 10^{0.5} \approx 3.16$ never reached this step size and still performed well.

## 4.4 Theoretical considerations

Usually, convergence in deep learning is shown for convex stochastic functions with a L-Lipschitz continuous gradient. However, since our approach originates from empirical results, it is not given that a profound theoretical analysis is possible. In order to show any convergence guarantees for parabolic approximations, we have to fall back to uncommonly strong assumptions which lead to quadratic models. Since convergence proofs on quadratics are of minor importance for most readers, our derivations can be found in Appendix C.

## 5 Evaluation

### 5.1 Experimental design

We performed a comprehensive evaluation to analyze the performance of *PAL* on a variety of deep learning optimization tasks. Therefore, we tested *PAL* on commonly used architectures on CIFAR-10 [31], CIFAR-100 [31] and ImageNet [13]. For CIFAR-10 and CIFAR-100, we evaluated on DenseNet40 [25], EfficientNetB0 [56], ResNet32 [23] and MobileNetV2 [52]. On ImageNet we evaluated on DenseNet121 and ResNet50. In addition, we considered an RNN trained on the Tolstoi war and peace text prediction task. We compare *PAL* to *SLS* [58], whose Armijo variant is state-of-the-art in the line search field for DNNs. In addition, we compare against the following well studied and widely used first order optimizers: *SGD* with momentum [49], *ADAM* [30], and *RMSProp* [57] as well as against *SGDHD* [3], *ALIG* [5], which automatically estimate learning rates in negative gradient direction and, finally, against the coin betting approach *COCOB* [42]. To perform a fair comparison, we compared a variety of hyperparameter combinations of commonly used hyperparameters for each optimizer. In addition, we utilize those combinations to analyze the hyperparameter sensitivity for each optimizer. Since a grid search on Imagenet was too expensive, the best hyperparameter configuration from the CIFAR-100 evaluation was used to test hyperparameter transferability. A detailed explanation of the experiments including hyperparameters and data augmentations used are given in Appendix D.8. All in all, we trained over 4500 networks with Tensorflow 1.15 [1] on Nvidia Geforce GTX 1080 TI graphic cards. Since *PAL* is a line search approach, the predefined learning rate schedules of SGD and the generated schedules of *SLS*, *ALIG*, *SGDHD* and *PAL* were compared. Due to normalization, *PAL*'s learning rate is given by $s_{upd_t}/||d_t||$.

### 5.2 Results

A selection of our results is given in Figure 5. The results of other architectures trained on CIFAR-10, CIFAR-100, Imagenet and Tolstoi are found in Appendix D Figures 13,14,15. A table with exact numerical results of all experiments is provided in Appendix D.9.

In most cases *PAL* decreases the training loss faster and to a lower value than the other optimizers (row 1 of Figures 5,13,14,15). Considering validation and test accuracy, *PAL* surpasses *ALIG*, *SGDHD* and *COCOB*, competes with *RMSProp* and *ADAM* but gets surpassed by *SGD* (rows 2,3 of Figures 5,13,14,15). However, *RMSProp*, *ADAM* and *SGD* were tuned with a step size schedule. If we compare *PAL* to their basic implementations without a schedule, which roughly corresponds to the first plateau reached in row 2 of Figures 5,13,14,15, *PAL* would surpass the other optimizers and shows that it can find a well performing step size schedule. This is especially interesting for problems for which default schedules might not work.

*SLS* decreases the training loss further than the other optimizers on a few problems, but shows weak performance and poor generalization on most. This contrasts to the results of [58], where *SLS* behaves robustly and excels. To exclude the possibility of errors on our side, we reimplemented *SLS* experiment on ResNet34 and could reproduce a similar well performance as in [58] (Appendix D.3). Our results suggest, that the interpolation assumption on which *SLS* is based, is not always valid for the considered tasks.

Considering the box plots of Figures 5 and 14, which represent the sensitivity to hyperparameter combinations, one would likely try on a new unknown objective, we can see, that *PAL* has a strong tendency to exhibit low sensitivity in combination with good performance. To emphasize this statement, a sensitivity analysis of *PAL*'s hyperparameters (Appendix Figure 19) shows that *PAL* performs well on a wide range for each hyperparameter on a ResNet32.

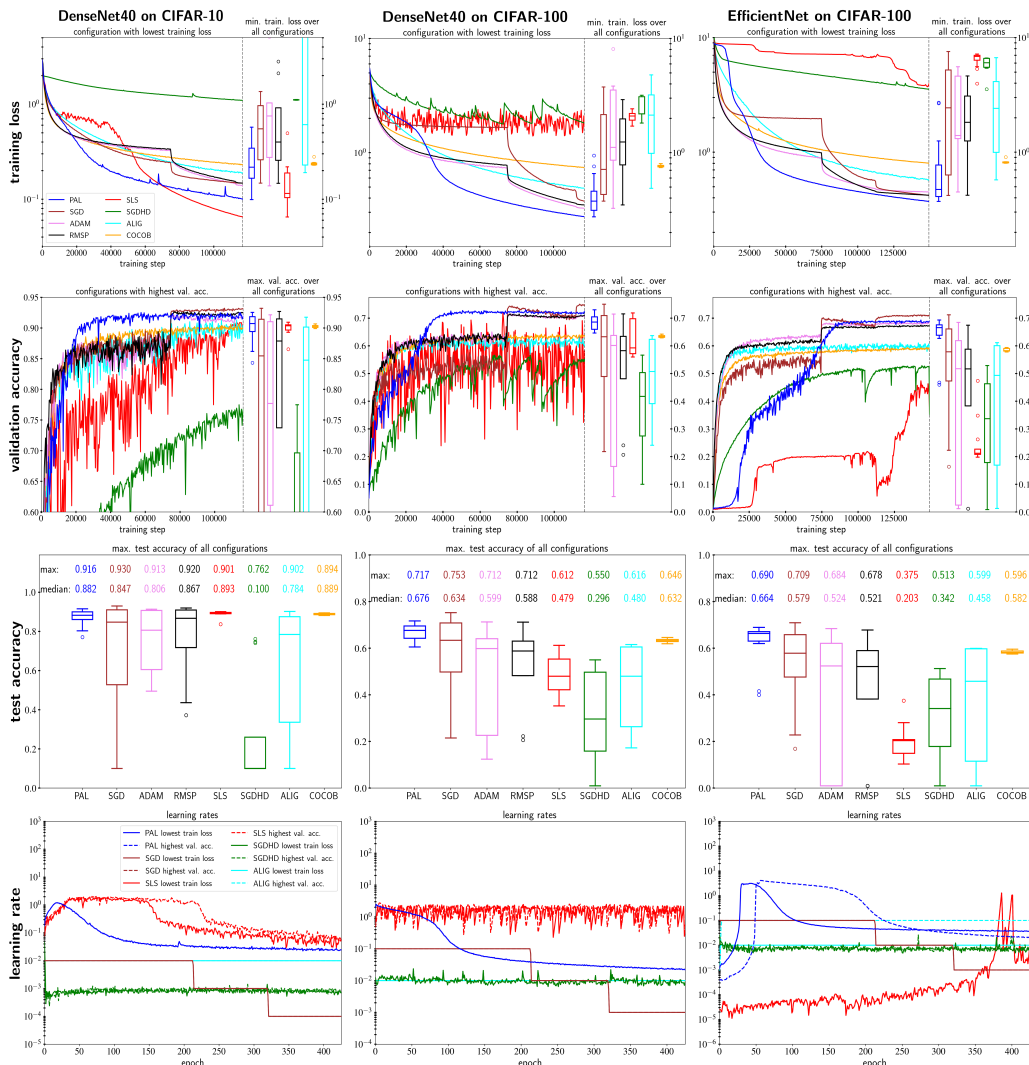

Figure 5: Comparison of *PAL* against *SLS*, *SGD*, *ADAM*, *RMSProp*, *ALIG*, *SGDHD* and *COCOB* on train. loss (row 1), val. acc. (row 2), test. acc. (row 3) and *SLS*, *SGD*, *ALIG*, *SGDHD* and *PAL* on learning rates (row 4). Comparison is done across several datasets and models. Further results are found in Appendix D.1 Figure (13,14,15). Results are averaged over 3 runs. Box plots result from comprehensive hyperparameter grid searches in plausible intervals. Learning rates are averaged over epochs. *PAL* surpasses, *ALIG*, *SGDHD*, and *COCOB* and competes against all other optimizers except against SGD.

On wall-clock-time *PAL* performs as fast as *SLS* but slower than the other optimizers, which achieve similar speeds (Appendix D.2). However, depending on the scenario, an automatic, well performing leaning rate schedule might compensate for the slower speed.

Considering the learning rate schedules of *PAL* (row 4 of Figures 5,13,14,15) we achieved unexpected results. *PAL*, which estimates the learning rate directly from approximated local shape information, does not follow a schedule that is similar to the one of *SLS*, *ALIG*, *SGDHD* or any of the common used hand crafted schedules such as piece wise constant or cosine decay. However, it achieves similar results. An interesting side result is that *ALIG* and *SGDHD* tend to perform best, if hyperparameters are chosen in a way that the learning rate is only changed slightly and therefore virtually an SGD training with fixed learning rate is performed.

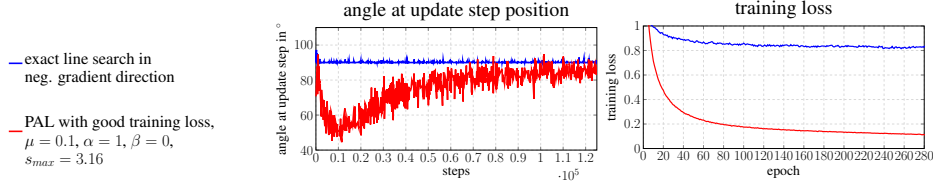

Figure 6: Comparison of *PAL* against an expensive exact line search. The first plot shows the angle between the direction and gradient vector at the update step position. A ResNet32 was trained on CIFAR-10. One can observe that an exact line search exhibits poor performance.

# 6 On the exactness of line searches on batch losses

In this section we investigate the general question whether line searches which estimate the location of the minimum of batch losses exactly are beneficial. In Figure 2 we showed that *PAL* can perform an almost exact line search on batch losses if we use a fixed update step adaptation factor (Section 4.3). However, *PAL*'s best hyperparameter configuration does not perform an exact line search (see Figure 6). Consequently, we analyzed how an exact line search, which exactly estimates a minimum of the line function, behaves. We implemented an inefficient binary line search (see Appendix E), which measured up to 20 values on each line to estimate the position of a minimum. The results, given in Figure 6, show that an optimal line search does not optimize well. Thus, the reason why *PAL* performs well is not the exactness of its update steps. In fact, slightly inexact update steps seem to be beneficial.

These results query Assumption 1, which assumes that the position of a minimum on a line in negative gradient direction of the batch loss $L(\mathbf{x}_t; \theta)$ is a suitable estimator for the minimum of the empirical loss $\mathcal{L}$ on this line to perform a successful optimization process. To investigate this further, we tediously measured the empirical loss $\mathcal{L}$ and the distribution of batch losses for one training process on a ResNet32. Our results suggest, as exemplary shown in Figure 7, that on a line function defined by the gradient of $L(\mathbf{x}_t; \theta)$, the position of the minimum of $L(\mathbf{x}_t; \theta)$ is not always a good estimator for the position of the minimum of the empirical loss $\mathcal{L}$. This explains why exact line searches on the batch loss perform weak.

Corollaries are that the empirical loss on the investigated lines also tends to be locally convex and that the optimal step size tends to be smaller than the step size given by the batch loss on such lines. This is a possible explanation why the slightly too narrow parabolic approximations of *PAL* without update step adaptation perform well.

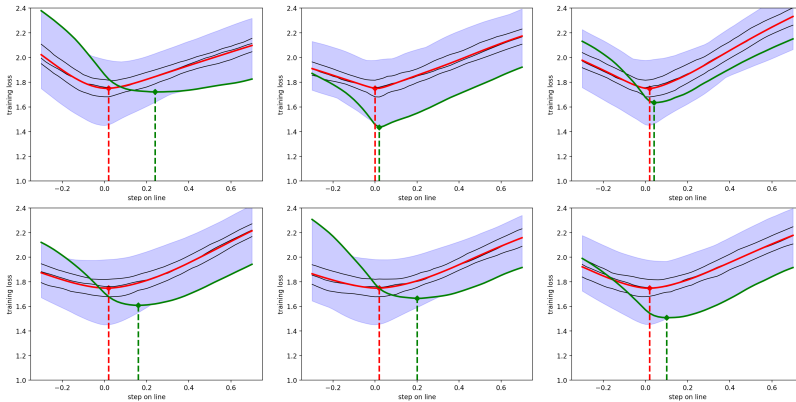

Figure 7: Distributions (blue) over all batch losses on representative cross sections during a training of a ResNet32 on CIFAR-10. The empirical loss, which is the mean value of the distribution, is given in red. The quartiles are given in black. The batch loss, whose negative gradient defines the search direction, is given in green. It can be observed that the minimum of the green batch loss is not always an adequate estimator of the minimum of the empirical loss on the corresponding cross section.

# 7 PAL and Interpolation

This section analyzes whether the reason why *PAL* performs well is related to the interpolation condition. Formally, interpolation requires that the gradient with respect to each sample converges to zero at the optimum. We repeated the experiments of the *SLS* paper (see [58] Section 7.2 and 7.3), which analyze the performance on problems for which interpolation hold or does not hold.

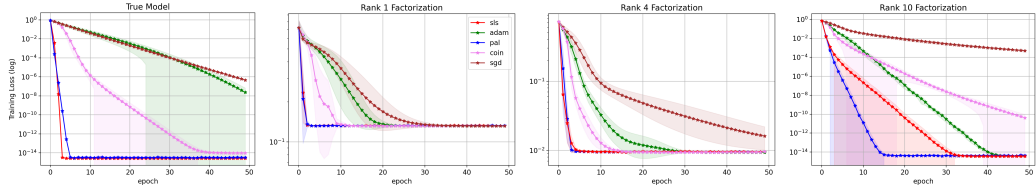

Figure 8: The matrix factorization problem of [58] Section 7.2. For $k = 1$ and $k = 4$ interpolation does not hold. Rank 1 factorization is under-parameterized, whereas rank 4 and rank 10 factorizations are over-parameterized.

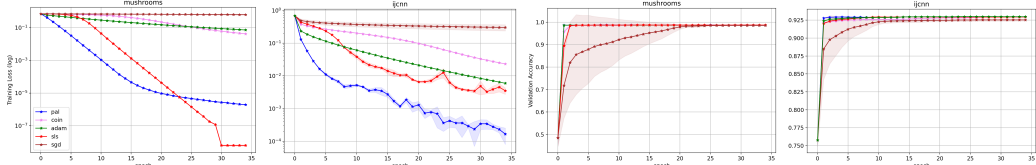

Figure 9: Binary classification task of [58] Section 7.3 using a softmax loss and RBF kernels for mushrooms and ijcnn datasets. With RBF kernels, the mushrooms dataset is linear separable in kernel-space with the selected kernel bandwidths, while the ijcnn dataset is not.

Figure 8 shows that *PAL* such as *SLS* converge faster to an artificial optimization floor on non-over-parameterized models ($k = 4$) of the matrix factorization problem of [58] Section 7.2. In the interpolation case *PAL* and *SLS* converge linearly to machine precision. On the binary classification problem of [58] Section 7.3, which uses a softmax loss and RBF kernels on the mushrooms and ijcnn datasets, we observe that *PAL* and *SLS* converge fast on the mushrooms task, for which the interpolation condition holds (Figure 9). However, *PAL* converges faster on the ijcnn task, for which the interpolation condition does not hold.

The results indicate that the interpolation condition is beneficial for *PAL*, but, *PAL* performs also robust when it is likely not satisfied (see Figure 5,13,14,15. In those experiments *PAL* mostly performs competitive but *SLS* does not. However, the relation of the parabolic property to interpolation needs to be investigated more closely in future.

# 8 Conclusions

This work tackles a major challenge in current optimization research for deep learning: to automatically find optimal step sizes for each update step. In detail, we focus on line search approaches to deal with this challenge. We introduced a simple, robust and competitive line search approach based on one-dimensional parabolic approximations of batch losses. The introduced algorithm is an alternative to *SGD* for objectives where default decays are unknown or do not work.

Loss functions of DNNs are commonly perceived as being highly non-convex. Our analysis suggests that this intuition does not hold locally, since lines of loss landscapes across models and datasets can be approximated parabolically to high accuracy. This new knowledge might further help to explain why update steps of specific optimizers perform well.

To gain deeper insights of line searches in general, we analyzed how an expensive but exact line search on batch losses behaves. Intriguingly, its performance is weak, which lets us conclude that the small inaccuracies of the parabolic approximations are beneficial for training.

## Potential Broader Impact

Since we understand our work as basic research, it is extremely error-prone to estimate its *specific* ethical aspects and future positive or negative social consequences. As optimization research influences the whole field of deep learning, we refer to the following works, which discuss the ethical aspects and social consequences of AI and Deep Learning in a comprehensive and general way: [6, 41, 61].

## Acknowledgments

Maximus Mutschler heartly thanks Lydia Federmann, Kevin Laube, Jonas Tebbe, Mario Laux, Valentin Bolz, Hauke Neitzel, Leon Varga, Benjamin Kiefer, Timon Höfer, Martin Meßmer, Cornelia Schulz, Hamd Riaz, Nuri Benbarka, Samuel Scherer, Frank Schneider, Robert Geirhos and Frank Hirschmann for their comprehensive support.

## Funding

This research was supported by the German Federal Ministry of Education and Research (BMBF) project 'Training Center Machine Learning, Tübingen' with grant number 01|S17054.

## Footnotes

[1]This holds because if the directional derivative of the measured gradient in line direction is 0, the current position is an extremum or saddle point of the cross sections and the angle is $90°$. If the position is not a extremum or saddle point, the directional derivative is not 0 [28].

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
