[Supplementary Material 1 · full_paper_with_appendix.pdf]

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

# A    Further line plots

Figure 10: **DenseNet40** loss functions on lines in negative gradient direction (blue) combined with our parabolic approximation (orange) and the position of the minimum (red). The unit of the horizontal axis is the change of $\theta$.

Figure 11: Loss line plots for **MobileNetV2**. For explanations see Figure 8. During training the parabolic approximation fits less accurately on the right hand side and during the first 150 steps it does not fit at all, however, the minimum of the parabola is still a good estimator for a low loss value on the line.

Figure 12: Loss line plots for **EfficientNet**. For explanations see Figure 8. The parabolic approximation fits only on the left hand side and during the first 150 steps it does not fit at all, however, the minimum of the parabola is still a good estimator for a low loss value on the line.

## B  PAL with all additions:

Algorithm 2 provides the full description of *PAL* including the additions described in Section 4.3. After analyzing the best hyperparameter combinations of *PAL* over all experiments, we suggest to use values from the following parameter intervals: $\mu = [0.1, 1]$, $\alpha = [1.0, 1.6]$, $\beta = [0, 0.4]$, $s_{max} = 3$. Where $\alpha, \beta$ and $s_{max}$ usually have a low sensitivity. Thus, the basic implementation of *PAL* (Algorithm 1) performs already well and is always found in the upper quartile considering the performance of all analyzed hyperparameter configurations in our experiments. PyTorch and Tensorflow 1.5 implementations are provided at `https://github.com/cogsys-tuebingen/PAL`.

---

**Algorithm 2** *PAL*, our proposed line search algorithm for DNNs. See Section 4 for details.

---

**Input:** Hyperparameters: $\mu$: measuring step size, $\alpha$: update step adaptation, $\beta$: direction adaptation factor, $s_{max}$: maximum step size.
**Input:** $L(x; \theta)$: loss function
**Input:** $x$: list of input vectors
**Input:** $\theta_0$: initial parameter vector
1: $t \leftarrow 0$
2: $d_t \leftarrow \vec{0}$
3: **while** $\theta_t$ not converged **do**
4: $\quad l_0 \leftarrow L(\mathbf{x}_t; \theta_t)$
5: $\quad d_t \leftarrow -\nabla_{\theta_t} L(\mathbf{x}_t; \theta_t) + \beta d_{-1}$
6: $\quad l_\mu \leftarrow L(\mathbf{x}_t; \theta_t + \mu \frac{d_t}{||d_t||})$
7: $\quad b \leftarrow \nabla_{\theta_t} L(\mathbf{x}_t; \theta_t) \frac{d_t}{||d_t||}$
8: $\quad a \leftarrow \frac{l_\mu - l_0 - b\mu}{\mu^2}$

9: $\quad$ **if** $a > 0$ **and** $b < 0$ **then**
10: $\quad\quad s_{upd} \leftarrow -\alpha \frac{b}{2a}$
11: $\quad$ **else if** $a \leq 0$ **and** $b < 0$ **then**
12: $\quad\quad s_{upd} \leftarrow \mu$
13: $\quad$ **else**
14: $\quad\quad s_{upd} \leftarrow 0$
15: $\quad$ **end if**
16: $\quad$ **if** $s_{upd} > s_{max}$ **then**
17: $\quad\quad s_{upd} \leftarrow s_{max}$
18: $\quad$ **end if**
19: $\quad \theta_{t+1} \leftarrow \theta_t + s_{upd} \frac{d_t}{||d_t||}$
20: $\quad t \leftarrow t + 1$
21: **end while**
22: **return** $\theta_t$

---

## C  Further Theoretical Considerations

We have to note, that the following derivation is based upon **strong assumptions** and if they are valid at all than they are likely more valid locally than globally. We assume that each slice of the loss function is a one-dimensional parabolic function:

**Assumption 2.** *Let $n \in \mathbb{N}$ be the number of parameters and let $\mathbf{l}, \mathbf{d} \in \mathbb{R}^n$ be vectors. Then for all $\mathbf{l}, \mathbf{d}$ there exists $a, b, c \in \mathbb{R}$ with $a > 0$, such that $L(\mathbf{x}_t; \mathbf{l} + \mathbf{d}s) = as^2 + bs + c$ for all $s \in \mathbb{R}$.*

This **strong** assumption is a simplified adaptation to our empirical results that lines in negative gradient direction behave locally almost parabolic (see Section 3). For the following derivations we assume a basic *PAL* without the additions introduced in Section 4.3. Proofs are provided in Appendix C.1. At first we show that $L(\mathbf{x}_t, \theta)$ is a n-dimensional parabolic function:

**Lemma 1.** *Let $f : \mathbb{R}^n \to \mathbb{R}$ be a k-times continuously differentiable function. Furthermore, assume there exists $a, b, c \in \mathbb{R}$ with $a > 0$, such that $f(\mathbf{l} + \mathbf{d}s) = as^2 + bs + c$ for all $s \in \mathbb{R}$. Then there exist $z \in \mathbb{R}, \mathbf{r} \in \mathbb{R}^n$ and a positive definite Matrix $\mathbf{Q} \in \mathbb{R}^{n \times n}$ such that $f(\mathbf{x}) = c + \mathbf{r}^T\mathbf{x} + \mathbf{x}^T\mathbf{Q}\mathbf{x}$ for all $\mathbf{x} \in \mathbb{R}^n$.*

Now we show that *PAL* converges on $L(\mathbf{x}_t, \theta)$:

**Proposition 1.** *PAL converges on $f(\mathbf{x}) : \mathbb{R}^n \to \mathbb{R}, f(\mathbf{x}) = c + \mathbf{r}^T\mathbf{x} + \mathbf{x}^T\mathbf{Q}\mathbf{x}$ with $\mathbf{Q} \in \mathbb{R}^{n \times n}$ hermitian and positive definite.*

We have to note, that in this scenario PAL is identical to the method of steepest decent for which the convergence including convergence rates on quadratics is already proven in [36] page 235. Nevertheless we have attached our own proof.

For a noisy scenario where each batch defines a quadratic, *PAL* has no convergence guarantee. Given two shifted one-dimensional parabolas, $ax^2 + bx + c$ and $a(x + d)^2 + b(x + d) + c$, which are

presented to *PAL* alternately, *PAL* will always perform an update step to the minimum position of one of these but never to the minimum position of the average of both. By slightly changing the training procedure and assuming that each $L(\mathbf{x_i}, \theta)$ has the same $\mathbf{Q}$ this can be fixed:

**Proposition 2.** *If $\mathcal{L}(\theta) : \mathbb{R}^n \to \mathbb{R} \; \theta \mapsto \mathcal{L}(\theta) = \frac{1}{m} \sum_{i=1}^m c_i + \mathbf{r}_i^T \theta + \theta^T \mathbf{Q}_i \theta$ and $c_i + \mathbf{r}_i^T \theta + \theta^T \mathbf{Q}_i \theta = L(\theta; \mathbf{x}_i)$ with $m$ being number the of batches and $\mathbf{x}_i$ defining one batch. (Each batch defines a parabola. The empirical loss $\mathcal{L}(\theta)$ is the mean of these parabolas). And for all $i, j \in \mathbb{N}$ it holds that $\mathbf{Q}_i = \mathbf{Q}_j$ and that $\mathbf{Q_i}$ is positive definite. Then $\arg \min_\theta \mathcal{L}(\theta) = \frac{1}{m} \sum_{i=1}^m \arg \min_\theta L(\theta)$ holds.*

This implies that under Assumption 2 and a fixed $\mathbf{Q}$ the position of the minimum of the empirical loss is given by the average of the positions of the minima of the batch losses. The minimum position of the empirical loss is found by *PAL*, by slightly adapting *PAL* to search on one batch until it finds the position of the minimum and then averaging the minima of each batch. As a result, *PAL* converges in this noisy scenario. However, we have to emphasize at this point that our assumptions about $\mathbf{l}$ and $\mathbf{Q}$ are likely not valid for general deep learning scenarios. But, if it is locally valid, this direction might be a further explanation, in addition to those of [17, 22], why stochastic weight averaging [27] performs well.

## C.1 Proofs

**Lemma 1.** *Let $f : \mathbb{R}^n \to \mathbb{R}$ a k-times continuously differentiable function. Furthermore, assume there exists $a, b, c \in \mathbb{R}$ with $a > 0$, such that $f(\mathbf{l} + \mathbf{d}s) = as^2 + bs + c$ for all $s \in \mathbb{R}$. Then there exist $z \in \mathbb{R}, \mathbf{r} \in \mathbb{R}^n$ and a positive definite Matrix $\mathbf{Q} \in \mathbb{R}^{n \times n}$ such that $f(\mathbf{x}) = c + \mathbf{r}^T \mathbf{x} + \mathbf{x}^T \mathbf{Q} \mathbf{x}$ for all $\mathbf{x} \in \mathbb{R}^n$.*

*Proof.*

$$g(\mathbf{x}) = u + \mathbf{v}^T \mathbf{x} + \mathbf{x}^T \mathbf{W} \mathbf{x} \text{ for some } u \in \mathbb{R}, \mathbf{v} \in \mathbb{R}^n \text{ and } \mathbf{W} \in \mathbb{R}^{n \times n}$$

$$\Leftrightarrow \forall \mathbf{l}, \mathbf{d} \in \mathbb{R}^n \wedge ||\mathbf{d}|| = 1 : \sum_{j=1}^n \sum_{k=1}^n \sum_{l=1}^n \frac{\partial^3 g(\mathbf{l})}{\partial x_j, \partial x_k, \partial x_l} d_j d_k d_l = 0 \tag{5}$$

$\Rightarrow$ holds since we have a polynomial of degree 2 and its third derivative is always a $\mathbf{0}$ tensor.
$\Leftarrow$ holds since the reminder of the quadratic Taylor expansion is always 0.

In our case the right part is 0 since:

$$\sum_{j=1}^n \sum_{k=1}^n \sum_{l=1}^n \frac{\partial^3 f(\mathbf{l})}{\partial x_j, \partial x_k, \partial x_l} d_j d_k d_l = \frac{\partial}{\partial s^3} f(\mathbf{l} + \mathbf{d}s) = 0 \tag{6}$$

In words: $f(\mathbf{x})$ is a parabolic function if and only if for each location $\mathbf{l}$ the third directional derivative of $f(\mathbf{l})$ in each direction $\mathbf{d}$ is 0. Which is the case, since the third derivative of each intersection is 0. $\mathbf{W}$ is positive definite since:

$$\forall \mathbf{d}, \mathbf{l} \in \mathbb{R}^n \wedge ||\mathbf{d}|| = 1 : \mathbf{d}^T \mathbf{W} \mathbf{d} = \frac{1}{2} \mathbf{d}^T \mathbf{H}(\mathbf{l}) \mathbf{d} = \frac{1}{2} \frac{\partial}{\partial s^2} f(\mathbf{l} + \mathbf{d}s) = a > 0 \tag{7}$$

where $\mathbf{H}$ is the Hessian. ∎

**Proposition 1.** *PAL converges on* $f(\mathbf{x}) : \mathbb{R}^n \to \mathbb{R}, f(\mathbf{x}) = c + \mathbf{r}^T\mathbf{x} + \mathbf{x}^T\mathbf{Q}\mathbf{x}$ *with* $\mathbf{Q} \in \mathbb{R}^{n \times n}$ *hermitian and positive definite.*

*Proof.*
For this prove we consider a basic *PAL* without the features introduced in Section 4.3. Note, that during the proof we will see, that $a > 0$ and $b < 0$. Thus, only the update step for this case has to be considered (see Section 4.2.

$f(\mathbf{x})$ is convex since $\mathbf{Q}$ is positive definite. Thus it has one minimum.
Without loss of generality we set $c = 0, \mathbf{r} = \mathbf{0}, \mathbf{x}_n \neq 0$

$$f(\mathbf{x}) = \mathbf{x}^T\mathbf{Q}\mathbf{x} \text{ and } \nabla_{\mathbf{x}}f(\mathbf{x}) = f'(\mathbf{x}) = 2\mathbf{Q}\mathbf{x} \qquad (8)$$

The values of $f(x)$ along a line through $\mathbf{x}$ in the direction of $-f'(\mathbf{x})$ are given by:

$$f(-f'(\mathbf{x})\hat{s} + \mathbf{x}) \qquad (9)$$

Now we expand the line function:

$$
\begin{aligned}
f(-f'(\mathbf{x})\hat{s} + \mathbf{x}) &= f(-2\mathbf{Q}\mathbf{x}\hat{s} + \mathbf{x}) \\
&= (-2\mathbf{Q}\mathbf{x}\hat{s} + \mathbf{x})^T\mathbf{Q}(-2\mathbf{Q}\mathbf{x}\hat{s} + \mathbf{x}) \\
&= \underbrace{4\mathbf{x}^T\mathbf{Q}^3\mathbf{x}}_{=:a}\,\hat{s}^2 + \underbrace{-4\mathbf{x}^T\mathbf{Q}^2\mathbf{x}}_{=:b}\,\hat{s} + \underbrace{\mathbf{x}^T\mathbf{Q}\mathbf{x}}_{=:c}
\end{aligned}
\qquad (10)
$$

Here we see that $f(\hat{s})$ is indeed a parabolic function with $a > 0$, $b < 0$ and $c > 0$ since $\mathbf{Q}^3$, $\mathbf{Q}^2$ and $\mathbf{Q}$ are positive definite.
The location of the minimum $s_{min}$ of $f(\hat{s})$ is given by:

$$\hat{s}_{min} = \arg\min_{\hat{s}} f(-f'(\mathbf{x})\hat{s} + \mathbf{x}) = -\frac{b}{2a} \qquad (11)$$

*PAL* determines $\hat{s}_{min}$ exactly with $\hat{s}_{min} = \frac{s_{upd}}{||f'(x)||}$ (see equation 1 and 2). $||f'(x)|| > 0$ since otherwise we are already in the minimum.

The value at the minimum is given by:

$$f(\hat{s}_{min}) = a(\frac{-b}{2a})^2 + b(\frac{-b}{2a}) + c = -\frac{b^2}{4a} + c = -\underbrace{\frac{(-\mathbf{x}^T\mathbf{Q}^2\mathbf{x})^2}{\mathbf{x}^T\mathbf{Q}^3\mathbf{x}}}_{=:g(\mathbf{x})} + \mathbf{x}^T\mathbf{Q}\mathbf{x} = -g(\mathbf{x}) + f(\mathbf{x}) \qquad (12)$$

Since $\mathbf{Q}^2$ and $\mathbf{Q}^3$ are positive definite and $\mathbf{x} \neq 0$:

$$g(\mathbf{x}) > 0 \qquad (13)$$

Now we consider the sequence $f(\mathbf{x}_n)$, with $\mathbf{x}_n$ defined by *PAL* (see Equation 1):

$$\mathbf{x}_{n+1} = -\frac{f'(\mathbf{x}_n)}{||f'(x_n)||}\hat{s}_{upd} + \mathbf{x}_n = -f'(\mathbf{x}_n)\hat{s}_{min} + \mathbf{x}_n \qquad (14)$$

It is easily seen by induction that:

$$0 < f(\mathbf{x}_{n+1}) < f(\mathbf{x}_n) = \sum_{i=0}^{n-1} -g(\mathbf{x}_i) + f(\mathbf{x}_0) < f(\mathbf{x}_0). \qquad (15)$$

$g(\mathbf{x}_n)$ converges to 0. Since $\forall n : g(\mathbf{x}_\mathbf{n}) > 0$ and $\sum_{i=0}^{n-1} -g(\mathbf{x}_i)$ is bounded.

Now we have to show that $\mathbf{x}_n$ converges to 0.
We have:

$$g(\mathbf{x}_n) = \frac{(\mathbf{x}_n^T\mathbf{Q}^2\mathbf{x}_n)^2}{\mathbf{x}_n^T\mathbf{Q}^3\mathbf{x}_n} = \frac{\langle \mathbf{x}_n, \mathbf{Q}^2\mathbf{x}_n \rangle^2}{\langle \mathbf{x}_n, \mathbf{Q}^3\mathbf{x}_n \rangle} \qquad (16)$$

Now we use the theorem of Courant-Fischer:

$$\langle x, x \rangle \min\{\lambda_1, \ldots, \lambda_n\} \leq \langle x, Ax \rangle \leq \langle x, x \rangle \max\{\lambda_1, \ldots, \lambda_n\} \tag{17}$$
$$\text{for any symmetric } A \in \mathbb{R}^{n \times n} \text{ with } \lambda_1, \ldots, \lambda_n$$

And get:

$$g(\mathbf{x}_n) \geq \frac{\lambda_{\mathbf{Q}^2 \min}^2 \langle \mathbf{x}_n, \mathbf{x}_n \rangle^2}{\lambda_{\mathbf{Q}^3 \max} \langle \mathbf{x}_n, \mathbf{x}_n \rangle} = C \frac{||\mathbf{x}_n||^4}{||\mathbf{x}_n||^2} = C||\mathbf{x}_n||^2 \tag{18}$$

with

$$C = \frac{\lambda_{\mathbf{Q}^2 \min}^2}{\lambda_{\mathbf{Q}^3 \max}} > 0 \text{ since all } \lambda \text{ of the positive definite } \mathbf{Q} \text{ are positive} \tag{19}$$

Thus, we have:

$$g(\mathbf{x}_n) \geq C||\mathbf{x}_n||^2 \geq 0 \tag{20}$$

Since $g(\mathbf{x}_n)$ converges to 0, $C||\mathbf{x}_n||^2$ converges to 0.
This means, $\mathbf{x}_n$ converges to $\mathbf{0}$, which is the location of the minimum. ∎

**Proposition 2.** *If $\mathcal{L}(\theta) : \mathbb{R}^n \to \mathbb{R} \; \theta \mapsto \mathcal{L}(\theta) = \frac{1}{m} \sum_{i=1}^m c_i + \mathbf{r}_i^T \theta + \theta^T \mathbf{Q}_i \theta$ and $c_i + \mathbf{r}_i^T \theta + \theta^T \mathbf{Q}_i \theta = L(\theta; \mathbf{x}_i)$ with $m$ being number the of batches and $\mathbf{x}_i$ defining one batch. (Each batch defines a parabola. The empirical loss $\mathcal{L}(\theta)$ is the mean of these parabolas). And for all $i, j \in \mathbb{N}$ it holds that $\mathbf{Q}_i = \mathbf{Q}_j$ and that $\mathbf{Q}_i$ is positive definite. Then $\arg \min_\theta \mathcal{L}(\theta) = \frac{1}{m} \sum_{i=1}^m \arg \min_\theta L(\theta)$ holds.*

*Proof.*
Since $\mathcal{L}(\theta)$ is a sum of convex functions, it is also convex and has one minimum.
At first we determine the derivative of $\mathcal{L}(\theta)$ with respect to $\theta$:

$$\frac{\partial}{\partial \theta} \mathcal{L}(\theta) = \frac{1}{m} \sum_{i=1}^m (\mathbf{r}_i + 2\mathbf{Q}_i \theta) = 2\mathbf{Q}\theta + \frac{1}{m} \sum_{i=1}^m \mathbf{r}_i \tag{21}$$

Then we determine the minima:

$$\arg \min_\theta \mathcal{L}(\theta) \Leftrightarrow \frac{\partial}{\partial \theta} \mathcal{L}(\theta) = \mathbf{0} \Leftrightarrow \theta = -\frac{1}{2} (\sum_{i=1}^m \mathbf{Q}_i)^{-1} \sum_{i=1}^m \mathbf{r}_i = -\frac{1}{2m} \mathbf{Q}^{-1} \sum_{i=1}^m \mathbf{r}_i \tag{22}$$

$$\arg \min_\mathbf{t} L(\mathbf{t} : \mathbf{x}_i) = -\frac{1}{2} \mathbf{Q}^{-1} \mathbf{r}_i \tag{23}$$

Thus, we get:

$$\arg \min_\theta \mathcal{L}(\theta) = -\frac{1}{2m} \mathbf{Q}^{-1} \sum_{i=1}^m \mathbf{r}_i = \frac{1}{m} \sum_{i=1}^m -\frac{1}{2} \mathbf{Q}^{-1} \mathbf{r}_i = \frac{1}{m} \sum_{i=1}^m \arg \min_t L(\mathbf{t} : \mathbf{x}_i) \tag{24}$$

∎

# D Further experimental results

## D.1 Performance Comparison on ImageNet, CIFAR-10, CIFAR-100 and Tolstoi

Figure 13: Comparison on **CIFAR-10** of *PAL* against *SLS*, *SGD*, *ADAM*, *RMSProp*, *ALIG*, *SGDHD* and *COCOB* on train. loss (row 1), val. acc. (row 2), test. acc. (row 3) and *SLS*, *SGD*, *ALIG*, *SGDHD* and *PAL* on learning rates (row 4). Results are averaged over 3 runs. Box plots result from comprehensive hyperparameter grid searches in plausible intervals. Learning rates are averaged over epochs. *PAL* surpasses *SLS*, *ALIG*, *SGDHD* and competes against all other optimizers except against SGD. The learning rate schedule comparison shows that *PAL* performs competitive although elaborating significantly different schedules.

Figure 14: Comparison on **CIFAR-100** of *PAL* against *SLS*, *SGD*, *ADAM*, *RMSProp*, *ALIG*, *SGDHD* and *COCOB* on train. loss (row 1), val. acc. (row 2), test. acc. (row 3) and *SLS*, *SGD*, *ALIG*, *SGDHD* and *PAL* on learning rates (row 4)). Results are averaged over 3 runs. Box plots result from comprehensive hyperparameter grid searches in plausible intervals. Learning rates are averaged over epochs. *PAL* surpasses *SLS*, *ALIG*, *SGDHD* and competes against all other optimizers except against SGD. The learning rate schedule comparison shows that *PAL* performs competitive although elaborating significantly different schedules.

Figure 15: Comparison of *PAL* to *SGD*, *SLS*, *ADAM*, *RMSProp* on training loss, validation accuracy and learning rates on **Imagenet**, and a simple RNN, trained on the **Tolstoi** War and Peace dataset. Learning rates are averaged over epochs. For Imagenet the best hyperparameter configuration from the CIFAR-100 evaluation were used to test hyperparameter transferability.

## D.2 Wall-clock time comparison

Table 1: Required seconds per epoch of *PAL*, *SLS*, *ALIG*, *SGDHD*, *COCOB* and *SGD* on CIFAR-10. RMSP and ADAM reach a similar speed as SGD. The comparison was performed on a Nvidia Geforce GTX 1080 TI. *PAL* and *SLS* perform slower, since they have to measure additional losses, whereas the additional operations of *ALIG*, *SGDHD*, *COCOB* tend to be cheap.

| network | seconds / epoch *PAL* | *SLS* | *SGD* | *ALIG* | *SGDHD* | *COCOB* |
|---|---|---|---|---|---|---|
| ResNet32 | 20.9 | 21.7 | 10.7 | 11.0 | 11.1 | 16.4 |
| MobilenetV2 | 53.2 | 52.4 | 34.1 | 34.01 | 34.2 | 36.6 |
| EfficientNet | 55.5 | 52.2 | 30.7 | 31.2 | 32.2 | 37.5 |
| DenseNet40 | 88.8 | 87.5 | 59.7 | 61.3 | 64.6 | 61.4 |

## D.3 SLS ResNet34 test case re-implementation

In the shown experiments and in contrast to the evaluation of *SLS* in [58], we used Tensorflow default Xavier weight initialization [19] versus PyTorch default Lecun initialization [33]. In addition, we used L2 regularisation versus no regularization. Furthermore, default implementations of networks for both frameworks have small differences. All in all those differences usually influence the optimizer performance only marginally as given by the fact that all other investigated optimizers perform well. However, in this case of *SLS* we see significant differences.

To prove that our implementation of *SLS* is correct, we re-implemented [58]'s ResNet34 test case on CIFAR-10 in Tensorflow and achieved similar results as [58]. SLS shows well performance and is not significantly overfitting as it does in in Section 5.2.

Figure 16: On the re-implemented ResNet34 test case of [58] SLS shows well performance and is not significantly overfitting as it does in in Section 5.2

## D.4 Parabolic property in adapted directions:

Figure 17: Angles between the line direction and the gradient at the estimated minimum measured on the same batch plotted over a whole training process on several networks on CIFAR-10. This figure clarifies, that parabolic property is also valid if a **direction adaptation factor of 0**.4 is applied. Measuring step sizes and update step adaptations factors (see Sections 4.1,4.3) were set to fit the cross sections decently.

## D.5 Influence of dynamic step sizes and the direction adaptation

This section analyses, whether *PAL*'s performance originates from dynamically chosen step sizes or from the the non-linear conjugate gradient like update step adaptation. We consider EfficientNets trained on CIFAR-10, since for those the update step adaptation factor $\beta$ is needed to achieve optimal results. We consider the following 6 scenarios: **1,2)** *PAL* without update step adaptation ($\beta = 0$) and with and without dynamic step sizes (Figure 18 left). **3,4)** *PAL* with a update step adaptation of $0.2$ and with and without dynamic step sizes (Figure 18 middle). **5,6)** *PAL* with a update step adaptation of $0.4$ with and without fixed step sizes (Figure 18 right). The case with fixed step sizes result in in normalized SGD (NSGD) with a momentum factor $\beta$. As fixed update step size we use the measuring step size $\mu$.

The results show that dynamic step sizes increase the performance always if direction adaptation is not applied and if it is applied in 6 out of 8 cases. Direction adaptation can increase or decrease the performance in both, the dynamic and the fixed step size cases. The best performance is achieved with a direction adaptation factor of $0.2$ and a measuring step size of $10^{-1.5}$, which shows that both factors influence the best results in this scenario.

Figure 18: Analysis of the influences of dynamic step sizes and the direction adaptation factor $\beta$.

## D.6 Sensitivity analysis:

All in all *PAL* tends to have a low hyperparameter sensitivity as shown in Figure 19. Since $\mu$ is the most sensitive hyperparameter we analyzed its sensitivity over several further models trained on CIFAR-10 (see Figure 20).

Figure 19: Sensitivity analysis for PAL on a ResNet32 trained on CIFAR-10. The baseline parameters are: $\mu = 0.1, \beta = 0.2, \alpha = 1.0, s_{max} = 10$. It shows that $\beta$ should be chosen $\leq 0.6$. $\alpha$ has a low sensitivity, but with a value of $1.4$ it reaches best performance. $s_{max}$ has a low sensitivity and all investigated values perform similarly. $\mu$ should be chosen between $10^{-2}$ and $10^{-0.5}$.

Figure 20: Sensitivity of the measuring step size $\mu$ of PAL for several models on CIFAR-10. *PAL* shows low sensitivity.

## D.7 Comparison to Probabilistic Line-Search (PLS):

We used a empirically improved and only existing implementation of *PLS* [38] for Tensorflow 1 [2]. However, the sum of squared gradients has to be derived manually for each layer, which is a considerable amount of work for modern architectures. Consequently, we limit our comparison to a ResNet-32 trained on CIFAR-10. Figure 21 shows that *PAL* and *PLS* perform similarly in this scenario.

Figure 21: Comparison of PAL to Probabilistic Line Search [38]

### D.8  Further experimental design details

#### D.8.1  Training Procedure

On CIFAR-10 and CIFAR-100 we trained 150k steps. On Imagenet each network was trained for 500k steps. We performed a piecewise constant learning rate decay by dividing the learning rate by 10 at 50% and 75% of the steps.

The training set to evaluation set split was 45k to 15k for CIFAR-10 and CIFAR-100. At the time of writing, the default Tensorflow classes do not support the reuse of the same randomly sampled numbers for multiple inferences, therefore, we implemented and used our own Dropout [55] layer.

To get a fair comparison of the optimizers capabilities, we compare on the training loss, the validation accuracy and the test accuracy metrics. For all metrics we provide the median and the quartiles to analyze the hyperparameter sensitivity. For each hyperparameter combination we averaged our results over 3 runs using the seeds 1, 2 and 3 for reproducibility. All in all, we trained over 4500 networks with Tensorflow 1.15 [1] on Nvidia Geforce GTX 1080 TI graphic cards.

#### D.8.2  Data Augmentation

On CIFAR-10 we performed the following augmentations [23]:
4 pixel padding and cropping, horizontal image flipping with probability 0.5.
On Imagenet we applied an initial random crop to 224x224 pixels. In addition, we applied lighting as described in [32]. For CIFAR-10, CIFAR-100 all images were normalized by channel-wise mean and variance. For the Tolstoi War and Peace dataset we omitted augmentation.

#### D.8.3  Hyperparameter grid search

For our evaluation we used all combinations out of the following common used hyperparameters. The batch size is always 128 except for DenseNets trained with *ALIG*, *SGDHD* and *COCOB* for which we encountered memory overflows and hat to reduce the batch size to 100. Weight decay is always $10^{-4}$.

On Imagenet, such a large grid search was not possible. In this case we compared with the best hyperparameter combinations found on Cifar-100.

*ADAM*:

| hyperparameter | symbol | values |
|---|---|---|
| learning rate | $\lambda$ | $\{1, 0.1, 0.01, 0.001, 0.0001\}$ |
| first momentum | $\beta_1$ | $\{0.9, 0.95\}$ |
| second momentum | $\beta_2$ | $\{0.999\}$ |
| epsilon | $\epsilon$ | $\{1e-8\}$ |

We did not vary the first or second momentum much, since [30] states that the values chosen are already good defaults.

*SGD*:

| hyperparameter | symbol | values |
|---|---|---|
| learning rate | $\lambda$ | $\{0.1, 0.01, 0.001, 0.0001\}$ |
| momentum | $\alpha$ | $\{0.85, 0.9, 0.95\}$ |

*RMSProp*:

| hyperparameter | symbol | values |
|---|---|---|
| learning rate | $\lambda$ | $\{0.1, 0.01, 0.001, 0.0001\}$ |
| discounting factor | $f$ | $\{0.9, 0.95\}$ |
| epsilon | $\epsilon$ | $\{1e-8\}$ |

*PAL*:

| hyperparameter | symbol | values |
|---|---|---|
| measuring step size | $\mu$ | $\{10^0, 10^{-0.5}, 10^{-0.1}, 10^{-0.15}\}$ |
| direction adaptation factor | $\beta$ | $\{0, 0.4\}$ |
| update step adaptation | $\alpha$ | $\{1, \frac{1}{0.8}\}$ |
| maximum step size | $s_{max}$ | $\{10^{0.5}(\approx 3.16)\}$ |

In our implementation we worked with a inverse update step adaptation $\gamma = \frac{1}{\alpha}$.

*SLS*:

| hyperparameter | symbol | values |
|---|---|---|
| initial step size | $\mu$ | $\{0.1, 1\}$ |
| step size decay | $\beta$ | $\{0.9, 0.99\}$ |
| step size reset | $\gamma$ | $\{2.0, 2.5\}$ |
| Armijo constant | $c$ | $\{0.1, 0.01\}$ |
| maximum step size | $\mu_{max}$ | $\{10.0\}$ |

*ALIG*:

| hyperparameter | symbol | values |
|---|---|---|
| maximal learning rate | $\lambda$ | $\{10, 1.0, 0.1, 0.01\}$ |
| momentum | $\beta$ | $\{0.85, 0.9, 0.95\}$ |

*COCOB*:

| hyperparameter | symbol | values |
|---|---|---|
| restriction factor | $\alpha$ | $\{25, 50, 75, 100, 125, 150, 175, 200\}$ |

*SGDHD*:

| hyperparameter | symbol | values |
|---|---|---|
| learning rate | $\lambda$ | $\{0.1, 0.01, 0.001\}$ |
| hyper gradient learning rate | $\beta$ | $\{0.1, 0.01, 0.001, 0.0001\}$ |

## D.9  Detailed numerical results

Table 2: Performance comparison of *PAL*, *RMSProp*, *ADAM*, *COCOB*, *SGDHD*, *ALIG* and *SGD*. All hyperparameter combinations given in Appendix D.8 were evaluated for each architecture. Results are averaged over 3 runs starting from different random seeds, except for training on ImageNet, for which results were not averaged. Note that tests on Imagenet were performed with the best hyperparameters found on CIFAR-100 to test the transferability of hyperparameters. Medians an Quartiles describe the distribution of results over reasonable hyper-parameter ranges.

| dataset | network | optimizer | training loss min | median; p25; p75 | validation accuracy max | median; p25; p75 | test accuracy max | median; p25; p75 |
|---|---|---|---|---|---|---|---|---|
| CIFAR-10 | EfficientNet | COCOB | 0.659 | 0.824; 0.739; 0.855 | 0.857 | 0.837; 0.832; 0.845 | 0.843 | 0.824; 0.818; 0.832 |
| | | ALIG | 0.279 | 0.89; 0.464; 1.911 | 0.906 | 0.805; 0.451; 0.895 | 0.893 | 0.757; 0.297; 0.878 |
| | | SGDHD | 2.002 | 6.239; 4.357; 7.803 | 0.834 | 0.657; 0.18; 0.74 | 0.828 | 0.647; 0.179; 0.731 |
| | | SLS | 2.837 | 5.596; 4.681; 6.292 | 0.653 | 0.357; 0.211; 0.443 | 0.643 | 0.357; 0.216; 0.442 |
| | | RMSP | 0.154 | 0.637; 0.333; 1.261 | **0.93** | 0.864; 0.658; 0.902 | 0.919 | 0.854; 0.648; 0.889 |
| | | ADAM | 0.155 | 0.818; 0.292; 2.275 | 0.926 | 0.841; 0.211; 0.907 | 0.919 | 0.83; 0.1; 0.896 |
| | | SGD | 0.165 | 2.287; 0.343; 4.221 | **0.93** | 0.872; 0.794; 0.915 | **0.921** | 0.862; 0.784; 0.906 |
| | | PAL | **0.137** | **0.244**; 0.186; 0.388 | 0.927 | **0.912**; 0.906; 0.921 | 0.916 | **0.902**; 0.889; 0.908 |
| CIFAR-10 | MobileNetV2 | COCOB | 0.232 | **0.282**; 0.257; 0.295 | 0.879 | 0.87; 0.866; 0.876 | 0.865 | 0.852; 0.848; 0.865 |
| | | ALIG | 0.183 | 0.938; 0.347; 1.926 | 0.914 | 0.695; 0.233; 0.888 | 0.897 | 0.528; 0.1; 0.851 |
| | | SGDHD | 0.698 | 2.234; 1.835; 4.366 | 0.886 | 0.75; 0.298; 0.807 | 0.877 | 0.737; 0.295; 0.791 |
| | | SLS | 1.387 | 2.462; 2.011; 2.584 | 0.667 | 0.443; 0.407; 0.504 | 0.595 | 0.4; 0.343; 0.437 |
| | | RMSP | **0.085** | 0.493; 0.337; 0.918 | 0.938 | 0.872; 0.675; 0.895 | 0.929 | 0.865; 0.664; 0.882 |
| | | ADAM | 0.095 | 0.477; 0.314; 1.861 | 0.939 | 0.874; 0.309; 0.896 | 0.93 | 0.864; 0.289; 0.886 |
| | | SGD | 0.149 | 0.878; 0.204; 1.552 | **0.947** | **0.907**; 0.87; 0.933 | **0.94** | **0.899**; 0.859; 0.925 |
| | | PAL | 0.15 | 0.377; 0.205; 0.531 | 0.92 | 0.905; 0.896; 0.91 | 0.905 | 0.886; 0.877; 0.896 |
| CIFAR-10 | DenseNet40 | COCOB | 0.228 | 0.234; 0.23; 0.24 | 0.907 | 0.903; 0.901; 0.904 | 0.894 | 0.889; 0.885; 0.892 |
| | | ALIG | 0.188 | 0.604; 0.227; 2.903 | 0.918 | 0.848; 0.438; 0.902 | 0.902 | 0.784; 0.336; 0.875 |
| | | SGDHD | 1.094 | 2.279; 1.349; 2.908 | 0.775 | 0.341; 0.099; 0.696 | 0.762 | 0.1; 0.1; 0.26 |
| | | SLS | **0.065** | **0.115**; 0.104; 0.189 | 0.91 | 0.904; 0.897; 0.905 | 0.901 | **0.893**; 0.89; 0.897 |
| | | RMSP | 0.147 | 0.398; 0.256; 0.915 | 0.927 | 0.879; 0.737; 0.915 | 0.92 | 0.867; 0.717; 0.909 |
| | | ADAM | 0.138 | 0.749; 0.274; 1.028 | 0.922 | 0.777; 0.611; 0.91 | 0.913 | 0.806; 0.605; 0.907 |
| | | SGD | 0.147 | 0.794; 0.396; 1.746 | **0.932** | 0.855; 0.537; 0.914 | **0.93** | 0.847; 0.528; 0.91 |
| | | PAL | 0.099 | 0.217; 0.165; 0.343 | 0.925 | **0.907**; 0.894; 0.919 | 0.916 | 0.882; 0.861; 0.9 |
| CIFAR-10 | ResNet32 | COCOB | 0.125 | 0.128; 0.127; 0.129 | 0.888 | 0.886; 0.885; 0.887 | 0.878 | 0.872; 0.871; 0.874 |
| | | ALIG | 0.122 | 0.658; 0.279; 1.485 | 0.892 | 0.815; 0.47; 0.881 | 0.866 | 0.71; 0.367; 0.852 |
| | | SGDHD | 0.35 | 0.464; 0.413; 0.701 | 0.864 | 0.835; 0.791; 0.843 | 0.837 | 0.796; 0.766; 0.827 |
| | | SLS | **0.005** | **0.006**; 0.005; 0.827 | 0.871 | 0.856; 0.758; 0.869 | 0.846 | 0.824; 0.657; 0.839 |
| | | RMSP | 0.105 | 0.199; 0.129; 0.498 | 0.922 | 0.884; 0.804; 0.904 | 0.915 | 0.877; 0.792; 0.896 |
| | | ADAM | 0.105 | 0.332; 0.133; 1.004 | 0.917 | 0.875; 0.677; 0.881 | 0.914 | 0.868; 0.654; 0.873 |
| | | SGD | 0.098 | 0.131; 0.118; 0.322 | **0.939** | **0.899**; 0.85; 0.924 | **0.933** | **0.893**; 0.838; 0.92 |
| | | PAL | 0.05 | 0.105; 0.075; 0.195 | 0.921 | 0.893; 0.887; 0.906 | 0.903 | 0.88; 0.849; 0.888 |
| CIFAR-100 | DenseNet40 | COCOB | 0.739 | 0.761; 0.75; 0.772 | 0.642 | 0.633; 0.631; 0.637 | 0.646 | 0.632; 0.629; 0.637 |
| | | ALIG | 0.488 | 2.125; 0.988; 3.128 | 0.637 | 0.508; 0.391; 0.623 | 0.616 | 0.48; 0.264; 0.605 |
| | | SGDHD | 1.78 | 2.6; 2.179; 3.465 | 0.566 | 0.418; 0.274; 0.504 | 0.55 | 0.296; 0.159; 0.497 |
| | | SLS | 1.367 | 1.908; 1.446; 1.96 | 0.719 | 0.593; 0.572; 0.698 | 0.612 | 0.479; 0.422; 0.554 |
| | | RMSP | 0.348 | 1.238; 0.78; 1.972 | 0.716 | 0.583; 0.481; 0.634 | 0.712 | 0.588; 0.482; 0.631 |
| | | ADAM | 0.326 | 1.114; 0.859; 3.53 | 0.715 | 0.601; 0.165; 0.637 | 0.712 | 0.599; 0.226; 0.641 |
| | | SGD | 0.376 | 0.713; 0.431; 2.154 | **0.75** | 0.633; 0.489; 0.709 | **0.753** | 0.634; 0.498; 0.708 |
| | | PAL | **0.275** | **0.376**; 0.312; 0.459 | 0.73 | **0.686**; 0.66; 0.705 | 0.717 | **0.676**; 0.642; 0.695 |
| CIFAR-100 | EfficientNet | COCOB | 0.802 | 0.817; 0.807; 0.822 | 0.594 | 0.583; 0.581; 0.59 | 0.596 | 0.582; 0.58; 0.588 |
| | | ALIG | 0.57 | 2.4; 0.995; 4.085 | 0.612 | 0.494; 0.169; 0.6 | 0.599 | 0.458; 0.115; 0.597 |
| | | SGDHD | 3.545 | 6.528; 5.519; 8.917 | 0.529 | 0.337; 0.178; 0.463 | 0.513 | 0.342; 0.179; 0.468 |
| | | SLS | 3.731 | 6.713; 6.348; 6.857 | 0.474 | 0.212; 0.208; 0.227 | 0.375 | 0.203; 0.149; 0.208 |
| | | RMSP | 0.422 | 1.823; 1.253; 2.968 | 0.675 | 0.517; 0.383; 0.588 | 0.678 | 0.521; 0.382; 0.59 |
| | | ADAM | 0.45 | 1.394; 1.312; 4.606 | 0.684 | 0.518; 0.025; 0.619 | 0.684 | 0.524; 0.01; 0.621 |
| | | SGD | 0.42 | 2.44; 0.633; 5.214 | **0.712** | 0.579; 0.473; 0.661 | **0.709** | 0.579; 0.476; 0.658 |
| | | PAL | **0.372** | **0.471**; 0.409; 0.772 | 0.693 | **0.666**; 0.638; 0.676 | 0.69 | **0.664**; 0.63; 0.671 |
| CIFAR-100 | MobileNetV2 | COCOB | 0.486 | **0.513**; 0.492; 0.536 | 0.644 | 0.63; 0.626; 0.638 | 0.644 | 0.63; 0.623; 0.638 |
| | | ALIG | 0.323 | 2.396; 0.817; 4.247 | 0.661 | 0.41; 0.034; 0.623 | 0.652 | 0.229; 0.01; 0.602 |
| | | SGDHD | 1.485 | 3.307; 2.425; 7.002 | 0.593 | 0.476; 0.39; 0.545 | 0.589 | 0.456; 0.385; 0.525 |
| | | SLS | 3.857 | 5.086; 5.031; 5.64 | 0.332 | 0.2; 0.099; 0.203 | 0.197 | 0.081; 0.052; 0.126 |
| | | RMSP | 0.198 | 1.518; 0.718; 3.368 | 0.728 | 0.593; 0.43; 0.635 | 0.727 | 0.593; 0.431; 0.634 |
| | | ADAM | 0.218 | 1.873; 0.776; 4.524 | 0.729 | 0.528; 0.025; 0.593 | 0.729 | 0.533; 0.02; 0.595 |
| | | SGD | 0.4 | 0.974; 0.473; 2.151 | **0.733** | 0.657; 0.57; 0.7 | **0.736** | 0.659; 0.573; 0.701 |
| | | PAL | **0.181** | 0.602; 0.314; 1.571 | 0.726 | **0.666**; 0.574; 0.689 | 0.722 | **0.664**; 0.509; 0.681 |
| CIFAR-100 | ResNet32 | COCOB | 0.498 | 0.569; 0.524; 0.673 | 0.609 | 0.608; 0.607; 0.608 | 0.605 | 0.602; 0.599; 0.604 |
| | | ALIG | 0.537 | 1.932; 0.995; 3.572 | 0.597 | 0.491; 0.19; 0.58 | 0.587 | 0.414; 0.144; 0.549 |
| | | SGDHD | 0.881 | 1.359; 1.06; 1.772 | 0.601 | 0.539; 0.472; 0.586 | 0.599 | 0.517; 0.431; 0.571 |
| | | SLS | 2.62 | 2.808; 2.78; 2.82 | 0.399 | 0.388; 0.384; 0.392 | 0.363 | 0.305; 0.274; 0.33 |
| | | RMSP | 0.519 | 1.019; 0.807; 2.083 | 0.661 | 0.599; 0.455; 0.651 | 0.656 | 0.603; 0.455; 0.65 |
| | | ADAM | 0.402 | 1.772; 0.768; 3.038 | 0.659 | 0.513; 0.262; 0.564 | 0.658 | 0.519; 0.255; 0.567 |
| | | SGD | 0.375 | **0.474**; 0.4; 1.522 | **0.697** | 0.614; 0.494; 0.672 | **0.694** | 0.616; 0.502; 0.667 |
| | | PAL | **0.339** | 0.485; 0.369; 1.424 | 0.662 | **0.636**; 0.546; 0.652 | 0.663 | **0.621**; 0.512; 0.647 |

| | | | | | | | | |
|---|---|---|---|---|---|---|---|---|
| TOLSTOI | RNN | COCOB | 1.506 | 1.56; 1.533; 1.593 | 0.589 | 0.58; 0.573; 0.584 | 0.582 | 0.572; 0.566; 0.577 |
| | | ALIG | 1.501 | 1.562; 1.528; 1.766 | 0.591 | 0.579; 0.523; 0.586 | 0.584 | 0.571; 0.513; 0.577 |
| | | SGDHD | 2.282 | 2.433; 2.379; 2.445 | 0.375 | 0.338; 0.336; 0.348 | 0.369 | 0.334; 0.332; 0.344 |
| | | SLS | 3.128 | 3.149; 3.136; 3.156 | 0.169 | 0.159; 0.158; 0.165 | 0.168 | 0.158; 0.157; 0.164 |
| | | RMSP | **1.475** | **1.509**; 1.492; 1.556 | **0.599** | **0.591**; 0.579; 0.595 | **0.592** | **0.583**; 0.572; 0.587 |
| | | ADAM | 1.516 | 1.655; 1.596; 1.681 | 0.588 | 0.567; 0.55; 0.578 | 0.581 | 0.561; 0.543; 0.571 |
| | | SGD | 1.496 | 1.872; 1.56; 2.675 | 0.594 | 0.483; 0.278; 0.573 | 0.587 | 0.476; 0.275; 0.566 |
| | | PAL | 1.528 | 1.569; 1.547; 1.588 | 0.587 | 0.581; 0.577; 0.586 | 0.579 | 0.571; 0.556; 0.575 |
| Imagenet | ResNet50 | RMSP | 9.485 | – | 0.286 | – | 0.28 | – |
| | | ADAM | 1.863 | – | 0.562 | – | 0.559 | – |
| | | SLS | 3.808 | – | 0.286 | – | 0.069 | – |
| | | SGD | 1.123 | – | **0.656** | – | **0.65** | – |
| | | PAL | **0.773** | – | 0.608 | – | 0.608 | – |
| Imagenet | DenseNet121 | RMSP | 6.901 | – | 0.0 | – | 0.0 | – |
| | | ADAM | 6.901 | – | 0.001 | – | 0.0 | – |
| | | SLS | 7.768 | – | 0.001 | – | 0.001 | – |
| | | SGD | 3.308 | – | 0.458 | – | 0.452 | – |
| | | PAL | **1.228** | – | **0.617** | – | **0.611** | – |

# E   Binary Line Search

The optimal binary line search we compared *PAL* against. Since the line decreases in negative gradient direction, at first a extrapolation phase performs as many steps forward as the loss does not increase. Afterwards a binary search is performed. This approach is valid if the underlying line is convex. For simple readability we chose Python 3.6 syntax.

```
Input:max_num_of_search_steps                                      1
def binary_line_search(last_loss, step, counter, is_extrapolate):  2
    if counter == max_num_of_search_steps:                         3
        return last_loss                                           4
    counter += 1                                                   5
    if is_extrapolate:                                             6
        current_loss = do_step_on_line(step)                       7
        if current_loss < last_loss:                               8
            return binary_line_search(current_loss, step,counter,  9
                is_extrapolate)
        else:                                                      10
            is_extrapolate = False                                 11
            do_step_on_line(-step,get_loss=False)                  12
    if not is_extrapolate:                                         13
        loss_right = do_step_on_line(0.5*step, True)               14
        if loss_right < last_loss:                                 15
            return binary_line_search(loss_right,                  16
                0.5*step,counter, is_extrapolate)
        loss_left = do_step_on_line(-1*step, True)                 17
        if loss_left < last_loss:                                  18
            return binary_line_search(loss_left,                   19
                0.5*step,counter, is_extrapolate)
        do_step_on_line(0.5*step,get_loss=False)                   20
        if loss_right >= last_loss and loss_left >= last_loss:     21
            return binary_line_search(loss_left, 0.5*step,         22
                counter, is_extrapolate)
        else:                                                      23
            # this state is not possible                           24
```

[Supplementary Material 2]

# A   Further line plots

Figure 10: **DenseNet40** loss functions on lines in negative gradient direction (blue) combined with our parabolic approximation (orange) and the position of the minimum (red). The unit of the horizontal axis is the change of $\theta$.

Figure 11: Loss line plots for **MobileNetV2**. For explanations see Figure 8. During training the parabolic approximation fits less accurately on the right hand side and during the first 150 steps it does not fit at all, however, the minimum of the parabola is still a good estimator for a low loss value on the line.

Figure 12: Loss line plots for **EfficientNet**. For explanations see Figure 8. The parabolic approximation fits only on the left hand side and during the first 150 steps it does not fit at all, however, the minimum of the parabola is still a good estimator for a low loss value on the line.

# B PAL with all additions:

Algorithm 2 provides the full description of *PAL* including the additions described in Section 4.3. After analyzing the best hyperparameter combinations of *PAL* over all experiments, we suggest to use values from the following parameter intervals: $\mu = [0.1, 1]$, $\alpha = [1.0, 1.6]$, $\beta = [0, 0.4]$, $s_{max} = 3$. Where $\alpha, \beta$ and $s_{max}$ usually have a low sensitivity. Thus, the basic implementation of *PAL* (Algorithm 1) performs already well and is always found in the upper quartile considering the performance of all analyzed hyperparameter configurations in our experiments. PyTorch and Tensorflow 1.5 implementations are provided at `https://github.com/cogsys-tuebingen/PAL`.

---

**Algorithm 2** *PAL*, our proposed line search algorithm for DNNs. See Section 4 for details.

---

**Input:** Hyperparameters: $\mu$: measuring step size, $\alpha$: update step adaptation, $\beta$: direction adaptation factor, $s_{max}$: maximum step size.
**Input:** $L(x; \theta)$: loss function
**Input:** $x$: list of input vectors
**Input:** $\theta_0$: initial parameter vector
1:   $t \leftarrow 0$
2:   $d_t \leftarrow \vec{0}$
3:   **while** $\theta_t$ not converged **do**
4:     $l_0 \leftarrow L(\mathbf{x}_t; \theta_t)$
5:     $d_t \leftarrow -\nabla_{\theta_t} L(\mathbf{x}_t; \theta_t) + \beta d_{-1}$
6:     $l_\mu \leftarrow L(\mathbf{x}_t; \theta_t + \mu \frac{d_t}{||d_t||})$
7:     $b \leftarrow \nabla_{\theta_t} L(\mathbf{x}_t; \theta_t) \frac{d_t}{||d_t||}$
8:     $a \leftarrow \frac{l_\mu - l_0 - b\mu}{\mu^2}$
9:     **if** $a > 0$ **and** $b < 0$ **then**
10:      $s_{upd} \leftarrow -\alpha \frac{b}{2a}$
11:    **else if** $a \leq 0$ **and** $b < 0$ **then**
12:      $s_{upd} \leftarrow \mu$
13:    **else**
14:      $s_{upd} \leftarrow 0$
15:    **end if**
16:    **if** $s_{upd} > s_{max}$ **then**
17:      $s_{upd} \leftarrow s_{max}$
18:    **end if**
19:    $\theta_{t+1} \leftarrow \theta_t + s_{upd} \frac{d_t}{||d_t||}$
20:    $t \leftarrow t + 1$
21: **end while**
22: **return** $\theta_t$

---

# C Further Theoretical Considerations

We have to note, that the following derivation is based upon **strong assumptions** and if they are valid at all than they are likely more valid locally than globally. We assume that each slice of the loss function is a one-dimensional parabolic function:

**Assumption 2.** *Let $n \in \mathbb{N}$ be the number of parameters and let $\mathbf{l}, \mathbf{d} \in \mathbb{R}^n$ be vectors. Then for all $\mathbf{l}, \mathbf{d}$ there exists $a, b, c \in \mathbb{R}$ with $a > 0$, such that $L(\mathbf{x}_t; \mathbf{l} + \mathbf{d}s) = as^2 + bs + c$ for all $s \in \mathbb{R}$.*

This **strong** assumption is a simplified adaptation to our empirical results that lines in negative gradient direction behave locally almost parabolic (see Section 3). For the following derivations we assume a basic *PAL* without the additions introduced in Section 4.3. Proofs are provided in Appendix C.1. At first we show that $L(\mathbf{x}_t, \theta)$ is a n-dimensional parabolic function:

**Lemma 1.** *Let $f : \mathbb{R}^n \to \mathbb{R}$ be a k-times continuously differentiable function. Furthermore, assume there exists $a, b, c \in \mathbb{R}$ with $a > 0$, such that $f(\mathbf{l} + \mathbf{d}s) = as^2 + bs + c$ for all $s \in \mathbb{R}$. Then there exist $z \in \mathbb{R}, \mathbf{r} \in \mathbb{R}^n$ and a positive definite Matrix $\mathbf{Q} \in \mathbb{R}^{n \times n}$ such that $f(\mathbf{x}) = c + \mathbf{r}^T \mathbf{x} + \mathbf{x}^T \mathbf{Q} \mathbf{x}$ for all $\mathbf{x} \in \mathbb{R}^n$.*

Now we show that *PAL* converges on $L(\mathbf{x}_t, \theta)$:

**Proposition 1.** *PAL converges on $f(\mathbf{x}) : \mathbb{R}^n \to \mathbb{R}, f(\mathbf{x}) = c + \mathbf{r}^T \mathbf{x} + \mathbf{x}^T \mathbf{Q} \mathbf{x}$ with $\mathbf{Q} \in \mathbb{R}^{n \times n}$ hermitian and positive definite.*

We have to note, that in this scenario PAL is identical to the method of steepest decent for which the convergence including convergence rates on quadratics is already proven in [36] page 235. Nevertheless we have attached our own proof.

For a noisy scenario where each batch defines a quadratic, *PAL* has no convergence guarantee. Given two shifted one-dimensional parabolas, $ax^2 + bx + c$ and $a(x + d)^2 + b(x + d) + c$, which are

presented to *PAL* alternately, *PAL* will always perform an update step to the minimum position of one of these but never to the minimum position of the average of both. By slightly changing the training procedure and assuming that each $L(\mathbf{x_i}, \theta)$ has the same $\mathbf{Q}$ this can be fixed:

**Proposition 2.** *If $\mathcal{L}(\theta) : \mathbb{R}^n \to \mathbb{R} \ \theta \mapsto \mathcal{L}(\theta) = \frac{1}{m} \sum_{i=1}^m c_i + \mathbf{r}_i^T \theta + \theta^T \mathbf{Q}_i \theta$ and $c_i + \mathbf{r}_i^T \theta + \theta^T \mathbf{Q}_i \theta = L(\theta; \mathbf{x}_i)$ with $m$ being number the of batches and $\mathbf{x}_i$ defining one batch. (Each batch defines a parabola. The empirical loss $\mathcal{L}(\theta)$ is the mean of these parabolas). And for all $i, j \in \mathbb{N}$ it holds that $\mathbf{Q}_i = \mathbf{Q}_j$ and that $\mathbf{Q_i}$ is positive definite. Then $\arg \min_{\theta} \mathcal{L}(\theta) = \frac{1}{m} \sum_{i=1}^m \arg \min_{\theta} L(\theta)$ holds.*

This implies that under Assumption 2 and a fixed $\mathbf{Q}$ the position of the minimum of the empirical loss is given by the average of the positions of the minima of the batch losses. The minimum position of the empirical loss is found by *PAL*, by slightly adapting *PAL* to search on one batch until it finds the position of the minimum and then averaging the minima of each batch. As a result, *PAL* converges in this noisy scenario. However, we have to emphasize at this point that our assumptions about l and $\mathbf{Q}$ are likely not valid for general deep learning scenarios. But, if it is locally valid, this direction might be a further explanation, in addition to those of [17, 22], why stochastic weight averaging [27] performs well.

## C.1   Proofs

**Lemma 1.** *Let $f : \mathbb{R}^n \to \mathbb{R}$ a k-times continuously differentiable function. Furthermore, assume there exists $a, b, c \in \mathbb{R}$ with $a > 0$, such that $f(\mathbf{l} + \mathbf{d}s) = as^2 + bs + c$ for all $s \in \mathbb{R}$. Then there exist $z \in \mathbb{R}, \mathbf{r} \in \mathbb{R}^n$ and a positive definite Matrix $\mathbf{Q} \in \mathbb{R}^{n \times n}$ such that $f(\mathbf{x}) = c + \mathbf{r}^T \mathbf{x} + \mathbf{x}^T \mathbf{Q} \mathbf{x}$ for all $\mathbf{x} \in \mathbb{R}^n$.*

*Proof.*

$$g(\mathbf{x}) = u + \mathbf{v}^T \mathbf{x} + \mathbf{x}^T \mathbf{W} \mathbf{x} \text{ for some } u \in \mathbb{R}, \mathbf{v} \in \mathbb{R}^n \text{ and } \mathbf{W} \in \mathbb{R}^{n \times n}$$

$$\Leftrightarrow \forall \mathbf{l}, \mathbf{d} \in \mathbb{R}^n \wedge ||\mathbf{d}|| = 1 : \sum_{j=1}^n \sum_{k=1}^n \sum_{l=1}^n \frac{\partial^3 g(\mathbf{l})}{\partial x_j, \partial x_k, \partial x_l} d_j d_k d_l = 0 \tag{5}$$

$\Rightarrow$ holds since we have a polynomial of degree 2 and its third derivative is always a $\mathbf{0}$ tensor.
$\Leftarrow$ holds since the reminder of the quadratic Taylor expansion is always 0.

In our case the right part is 0 since:

$$\sum_{j=1}^n \sum_{k=1}^n \sum_{l=1}^n \frac{\partial^3 f(\mathbf{l})}{\partial x_j, \partial x_k, \partial x_l} d_j d_k d_l = \frac{\partial}{\partial s^3} f(\mathbf{l} + \mathbf{d}s) = 0 \tag{6}$$

In words: $f(\mathbf{x})$ is a parabolic function if and only if for each location l the third directional derivative of $f(\mathbf{l})$ in each direction $\mathbf{d}$ is 0. Which is the case, since the third derivative of each intersection is 0. $\mathbf{W}$ is positive definite since:

$$\forall \mathbf{d}, \mathbf{l} \in \mathbb{R}^n \wedge ||\mathbf{d}|| = 1 : \mathbf{d}^T \mathbf{W} \mathbf{d} = \frac{1}{2} \mathbf{d}^T \mathbf{H}(\mathbf{l}) \mathbf{d} = \frac{1}{2} \frac{\partial}{\partial s^2} f(\mathbf{l} + \mathbf{d}s) = a > 0 \tag{7}$$

where $\mathbf{H}$ is the Hessian.                                                                            ∎

**Proposition 1.** *PAL converges on* $f(\mathbf{x}) : \mathbb{R}^n \to \mathbb{R}, f(\mathbf{x}) = c + \mathbf{r}^T \mathbf{x} + \mathbf{x}^T \mathbf{Q} \mathbf{x}$ *with* $\mathbf{Q} \in \mathbb{R}^{n \times n}$ *hermitian and positive definite.*

*Proof.*
For this prove we consider a basic *PAL* without the features introduced in Section 4.3. Note, that during the proof we will see, that $a > 0$ and $b < 0$. Thus, only the update step for this case has to be considered (see Section 4.2.

$f(\mathbf{x})$ is convex since $\mathbf{Q}$ is positive definite. Thus it has one minimum.
Without loss of generality we set $c = 0, \mathbf{r} = \mathbf{0}, \mathbf{x}_n \neq 0$

$$f(\mathbf{x}) = \mathbf{x}^T \mathbf{Q} \mathbf{x} \text{ and } \nabla_{\mathbf{x}} f(\mathbf{x}) = f'(\mathbf{x}) = 2\mathbf{Q}\mathbf{x} \tag{8}$$

The values of $f(x)$ along a line through $\mathbf{x}$ in the direction of $-f'(\mathbf{x})$ are given by:

$$f(-f'(\mathbf{x})\hat{s} + \mathbf{x}) \tag{9}$$

Now we expand the line function:

$$\begin{aligned} f(-f'(\mathbf{x})\hat{s} + \mathbf{x}) &= f(-2\mathbf{Q}\mathbf{x}\hat{s} + \mathbf{x}) \\ &= (-2\mathbf{Q}\mathbf{x}\hat{s} + \mathbf{x})^T \mathbf{Q}(-2\mathbf{Q}\mathbf{x}\hat{s} + \mathbf{x}) \\ &= \underbrace{4\mathbf{x}^T \mathbf{Q}^3 \mathbf{x}}_{=:a} \hat{s}^2 + \underbrace{-4\mathbf{x}^T \mathbf{Q}^2 \mathbf{x}}_{=:b} \hat{s} + \underbrace{\mathbf{x}^T \mathbf{Q} \mathbf{x}}_{=:c} \end{aligned} \tag{10}$$

Here we see that $f(\hat{s})$ is indeed a parabolic function with $a > 0$, $b < 0$ and $c > 0$ since $\mathbf{Q}^3$, $\mathbf{Q}^2$ and $\mathbf{Q}$ are positive definite.
The location of the minimum $s_{min}$ of $f(\hat{s})$ is given by:

$$\hat{s}_{min} = \arg\min_{\hat{s}} f(-f'(\mathbf{x})\hat{s} + \mathbf{x}) = -\frac{b}{2a} \tag{11}$$

*PAL* determines $\hat{s}_{min}$ exactly with $\hat{s}_{min} = \frac{s_{upd}}{||f'(x)||}$ (see equation 1 and 2). $||f'(x)|| > 0$ since otherwise we are already in the minimum.

The value at the minimum is given by:

$$f(\hat{s}_{min}) = a(\frac{-b}{2a})^2 + b(\frac{-b}{2a}) + c = -\frac{b^2}{4a} + c = -\underbrace{\frac{(-\mathbf{x}^T \mathbf{Q}^2 \mathbf{x})^2}{\mathbf{x}^T \mathbf{Q}^3 \mathbf{x}}}_{=:g(\mathbf{x})} + \mathbf{x}^T \mathbf{Q} \mathbf{x} = -g(\mathbf{x}) + f(\mathbf{x}) \tag{12}$$

Since $\mathbf{Q}^2$ and $\mathbf{Q}^3$ are positive definite and $\mathbf{x} \neq 0$:

$$g(\mathbf{x}) > 0 \tag{13}$$

Now we consider the sequence $f(\mathbf{x}_n)$, with $\mathbf{x}_n$ defined by *PAL* (see Equation 1):

$$\mathbf{x}_{n+1} = -\frac{f'(\mathbf{x}_n)}{||f'(x_n)||}\hat{s}_{upd} + \mathbf{x}_n = -f'(\mathbf{x}_n)\hat{s}_{min} + \mathbf{x}_n \tag{14}$$

It is easily seen by induction that:

$$0 < f(\mathbf{x}_{n+1}) < f(\mathbf{x}_n) = \sum_{i=0}^{n-1} -g(\mathbf{x}_i) + f(\mathbf{x}_0) < f(\mathbf{x}_0). \tag{15}$$

$g(\mathbf{x}_n)$ converges to 0. Since $\forall n : g(\mathbf{x_n}) > 0$ and $\sum_{i=0}^{n-1} -g(\mathbf{x}_i)$ is bounded.

Now we have to show that $\mathbf{x}_n$ converges to 0.
We have:

$$g(\mathbf{x}_n) = \frac{(\mathbf{x}_n^T \mathbf{Q}^2 \mathbf{x}_n)^2}{\mathbf{x}_n^T \mathbf{Q}^3 \mathbf{x}_n} = \frac{\langle \mathbf{x}_n, \mathbf{Q}^2 \mathbf{x}_n \rangle^2}{\langle \mathbf{x}_n, \mathbf{Q}^3 \mathbf{x}_n \rangle} \tag{16}$$

Now we use the theorem of Courant-Fischer:

$$\langle x, x \rangle \min\{\lambda_1, \ldots, \lambda_n\} \leq \langle x, Ax \rangle \leq \langle x, x \rangle \max\{\lambda_1, \ldots, \lambda_n\}$$

$$\text{for any symmetric } A \in \mathbb{R}^{n \times n} \text{ with } \lambda_1, \ldots, \lambda_n \tag{17}$$

And get:

$$g(\mathbf{x}_n) \geq \frac{\lambda_{\mathbf{Q}^2 \min}^2 \langle \mathbf{x}_n, \mathbf{x}_n \rangle^2}{\lambda_{\mathbf{Q}^3 \max} \langle \mathbf{x}_n, \mathbf{x}_n \rangle} = C \frac{||\mathbf{x}_n||^4}{||\mathbf{x}_n||^2} = C||\mathbf{x}_n||^2 \tag{18}$$

with

$$C = \frac{\lambda_{\mathbf{Q}^2 \min}^2}{\lambda_{\mathbf{Q}^3 \max}} > 0 \text{ since all } \lambda \text{ of the positive definite } \mathbf{Q} \text{ are positive} \tag{19}$$

Thus, we have:

$$g(\mathbf{x}_n) \geq C||\mathbf{x}_n||^2 \geq 0 \tag{20}$$

Since $g(\mathbf{x}_n)$ converges to 0, $C||\mathbf{x}_n||^2$ converges to 0.
This means, $\mathbf{x}_n$ converges to $\mathbf{0}$, which is the location of the minimum. ∎

**Proposition 2.** *If $\mathcal{L}(\theta) : \mathbb{R}^n \to \mathbb{R} \; \theta \mapsto \mathcal{L}(\theta) = \frac{1}{m}\sum_{i=1}^m c_i + \mathbf{r}_i^T \theta + \theta^T \mathbf{Q}_i \theta$ and $c_i + \mathbf{r}_i^T \theta + \theta^T \mathbf{Q}_i \theta = L(\theta; \mathbf{x}_i)$ with $m$ being number the of batches and $\mathbf{x}_i$ defining one batch. (Each batch defines a parabola. The empirical loss $\mathcal{L}(\theta)$ is the mean of these parabolas). And for all $i, j \in \mathbb{N}$ it holds that $\mathbf{Q}_i = \mathbf{Q}_j$ and that $\mathbf{Q}_i$ is positive definite. Then $\arg\min_\theta \mathcal{L}(\theta) = \frac{1}{m}\sum_{i=1}^m \arg\min_\theta L(\theta)$ holds.*

*Proof.*
Since $\mathcal{L}(\theta)$ is a sum of convex functions, it is also convex and has one minimum.
At first we determine the derivative of $\mathcal{L}(\theta)$ with respect to $\theta$:

$$\frac{\partial}{\partial \theta}\mathcal{L}(\theta) = \frac{1}{m}\sum_{i=1}^m (\mathbf{r}_i + 2\mathbf{Q}_i \theta) = 2\mathbf{Q}\theta + \frac{1}{m}\sum_{i=1}^m \mathbf{r}_i \tag{21}$$

Then we determine the minima:

$$\arg\min_\theta \mathcal{L}(\theta) \Leftrightarrow \frac{\partial}{\partial \theta}\mathcal{L}(\theta) = \mathbf{0} \Leftrightarrow \theta = -\frac{1}{2}(\sum_{i=1}^m \mathbf{Q}_i)^{-1}\sum_{i=1}^m \mathbf{r}_i = -\frac{1}{2m}\mathbf{Q}^{-1}\sum_{i=1}^m \mathbf{r}_i \tag{22}$$

$$\arg\min_{\mathbf{t}} L(\mathbf{t} : \mathbf{x}_i) = -\frac{1}{2}\mathbf{Q}^{-1}\mathbf{r}_i \tag{23}$$

Thus, we get:

$$\arg\min_\theta \mathcal{L}(\theta) = -\frac{1}{2m}\mathbf{Q}^{-1}\sum_{i=1}^m \mathbf{r}_i = \frac{1}{m}\sum_{i=1}^m -\frac{1}{2}\mathbf{Q}^{-1}\mathbf{r}_i = \frac{1}{m}\sum_{i=1}^m \arg\min_t L(\mathbf{t} : \mathbf{x}_i) \tag{24}$$

∎

# D Further experimental results

## D.1 Performance Comparison on ImageNet, CIFAR-10, CIFAR-100 and Tolstoi

Figure 13: Comparison on **CIFAR-10** of *PAL* against *SLS*, *SGD*, *ADAM*, *RMSProp*, *ALIG*, *SGDHD* and *COCOB* on train. loss (row 1), val. acc. (row 2), test. acc. (row 3) and *SLS*, *SGD*, *ALIG*, *SGDHD* and *PAL* on learning rates (row 4). Results are averaged over 3 runs. Box plots result from comprehensive hyperparameter grid searches in plausible intervals. Learning rates are averaged over epochs. *PAL* surpasses *SLS*, *ALIG*, *SGDHD* and competes against all other optimizers except against SGD. The learning rate schedule comparison shows that *PAL* performs competitive although elaborating significantly different schedules.

Figure 14: Comparison on **CIFAR-100** of *PAL* against *SLS*, *SGD*, *ADAM*, *RMSProp*, *ALIG*, *SGDHD* and *COCOB* on train. loss (row 1), val. acc. (row 2), test. acc. (row 3) and *SLS*, *SGD*, *ALIG*, *SGDHD* and *PAL* on learning rates (row 4)). Results are averaged over 3 runs. Box plots result from comprehensive hyperparameter grid searches in plausible intervals. Learning rates are averaged over epochs. *PAL* surpasses *SLS*, *ALIG*, *SGDHD* and competes against all other optimizers except against SGD. The learning rate schedule comparison shows that *PAL* performs competitive although elaborating significantly different schedules.

Figure 15: Comparison of *PAL* to *SGD*, *SLS*, *ADAM*, *RMSProp* on training loss, validation accuracy and learning rates on **Imagenet**, and a simple RNN, trained on the **Tolstoi** War and Peace dataset. Learning rates are averaged over epochs. For Imagenet the best hyperparameter configuration from the CIFAR-100 evaluation were used to test hyperparameter transferability.

## D.2 Wall-clock time comparison

Table 1: Required seconds per epoch of *PAL*, *SLS*, *ALIG*, *SGDHD*, *COCOB* and *SGD* on CIFAR-10. RMSP and ADAM reach a similar speed as SGD. The comparison was performed on a Nvidia Geforce GTX 1080 TI. *PAL* and *SLS* perform slower, since they have to measure additional losses, whereas the additional operations of *ALIG*, *SGDHD*, *COCOB* tend to be cheap.

| network | seconds / epoch *PAL* | *SLS* | *SGD* | *ALIG* | *SGDHD* | *COCOB* |
|---|---|---|---|---|---|---|
| ResNet32 | 20.9 | 21.7 | 10.7 | 11.0 | 11.1 | 16.4 |
| MobilenetV2 | 53.2 | 52.4 | 34.1 | 34.01 | 34.2 | 36.6 |
| EfficientNet | 55.5 | 52.2 | 30.7 | 31.2 | 32.2 | 37.5 |
| DenseNet40 | 88.8 | 87.5 | 59.7 | 61.3 | 64.6 | 61.4 |

## D.3 SLS ResNet34 test case re-implementation

In the shown experiments and in contrast to the evaluation of *SLS* in [58], we used Tensorflow default Xavier weight initialization [19] versus PyTorch default Lecun initialization [33]. In addition, we used L2 regularisation versus no regularization. Furthermore, default implementations of networks for both frameworks have small differences. All in all those differences usually influence the optimizer performance only marginally as given by the fact that all other investigated optimizers perform well. However, in this case of *SLS* we see significant differences.

To prove that our implementation of *SLS* is correct, we re-implemented [58]'s ResNet34 test case on CIFAR-10 in Tensorflow and achieved similar results as [58]. SLS shows well performance and is not significantly overfitting as it does in in Section 5.2.

Figure 16: On the re-implemented ResNet34 test case of [58] SLS shows well performance and is not significantly overfitting as it does in in Section 5.2

## D.4 Parabolic property in adapted directions:

Figure 17: Angles between the line direction and the gradient at the estimated minimum measured on the same batch plotted over a whole training process on several networks on CIFAR-10. This figure clarifies, that parabolic property is also valid if a **direction adaptation factor of 0**.4 is applied. Measuring step sizes and update step adaptations factors (see Sections 4.1,4.3) were set to fit the cross sections decently.

## D.5 Influence of dynamic step sizes and the direction adaptation

This section analyses, whether *PAL*'s performance originates from dynamically chosen step sizes or from the the non-linear conjugate gradient like update step adaptation. We consider EfficientNets trained on CIFAR-10, since for those the update step adaptation factor $\beta$ is needed to achieve optimal results. We consider the following 6 scenarios: **1,2)** *PAL* without update step adaptation ($\beta = 0$) and with and without dynamic step sizes (Figure 18 left). **3,4)** *PAL* with a update step adaptation of $0.2$ and with and without dynamic step sizes (Figure 18 middle). **5,6)** *PAL* with a update step adaptation of $0.4$ with and without fixed step sizes (Figure 18 right). The case with fixed step sizes result in in normalized SGD (NSGD) with a momentum factor $\beta$. As fixed update step size we use the measuring step size $\mu$.

The results show that dynamic step sizes increase the performance always if direction adaptation is not applied and if it is applied in 6 out of 8 cases. Direction adaptation can increase or decrease the performance in both, the dynamic and the fixed step size cases. The best performance is achieved with a direction adaptation factor of $0.2$ and a measuring step size of $10^{-1.5}$, which shows that both factors influence the best results in this scenario.

Figure 18: Analysis of the influences of dynamic step sizes and the direction adaptation factor $\beta$.

## D.6 Sensitivity analysis:

All in all *PAL* tends to have a low hyperparameter sensitivity as shown in Figure 19. Since $\mu$ is the most sensitive hyperparameter we analyzed its sensitivity over several further models trained on CIFAR-10 (see Figure 20).

Figure 19: Sensitivity analysis for PAL on a ResNet32 trained on CIFAR-10. The baseline parameters are: $\mu = 0.1, \beta = 0.2, \alpha = 1.0, s_{max} = 10$. It shows that $\beta$ should be chosen $\leq 0.6$. $\alpha$ has a low sensitivity, but with a value of $1.4$ it reaches best performance. $s_{max}$ has a low sensitivity and all investigated values perform similarly. $\mu$ should be chosen between $10^{-2}$ and $10^{-0.5}$.

Figure 20: Sensitivity of the measuring step size $\mu$ of PAL for several models on CIFAR-10. *PAL* shows low sensitivity.

## D.7 Comparison to Probabilistic Line-Search (PLS):

We used a empirically improved and only existing implementation of *PLS* [38] for Tensorflow 1 [2]. However, the sum of squared gradients has to be derived manually for each layer, which is a considerable amount of work for modern architectures. Consequently, we limit our comparison to a ResNet-32 trained on CIFAR-10. Figure 21 shows that *PAL* and *PLS* perform similarly in this scenario.

Figure 21: Comparison of PAL to Probabilistic Line Search [38]

## D.8 Further experimental design details

### D.8.1 Training Procedure

On CIFAR-10 and CIFAR-100 we trained 150k steps. On Imagenet each network was trained for 500k steps. We performed a piecewise constant learning rate decay by dividing the learning rate by 10 at 50% and 75% of the steps.

The training set to evaluation set split was 45k to 15k for CIFAR-10 and CIFAR-100. At the time of writing, the default Tensorflow classes do not support the reuse of the same randomly sampled numbers for multiple inferences, therefore, we implemented and used our own Dropout [55] layer. To get a fair comparison of the optimizers capabilities, we compare on the training loss, the validation accuracy and the test accuracy metrics. For all metrics we provide the median and the quartiles to analyze the hyperparameter sensitivity. For each hyperparameter combination we averaged our results over 3 runs using the seeds 1, 2 and 3 for reproducibility. All in all, we trained over 4500 networks with Tensorflow 1.15 [1] on Nvidia Geforce GTX 1080 TI graphic cards.

### D.8.2 Data Augmentation

On CIFAR-10 we performed the following augmentations [23]:
4 pixel padding and cropping, horizontal image flipping with probability 0.5.
On Imagenet we applied an initial random crop to 224x224 pixels. In addition, we applied lighting as described in [32]. For CIFAR-10, CIFAR-100 all images were normalized by channel-wise mean and variance. For the Tolstoi War and Peace dataset we omitted augmentation.

### D.8.3 Hyperparameter grid search

For our evaluation we used all combinations out of the following common used hyperparameters. The batch size is always 128 except for DenseNets trained with *ALIG*, *SGDHD* and *COCOB* for which we encountered memory overflows and hat to reduce the batch size to 100. Weight decay is always $10^{-4}$.

On Imagenet, such a large grid search was not possible. In this case we compared with the best hyperparameter combinations found on Cifar-100.

*ADAM*:

| hyperparameter | symbol | values |
|---|---|---|
| learning rate | $\lambda$ | $\{1, 0.1, 0.01, 0.001, 0.0001\}$ |
| first momentum | $\beta_1$ | $\{0.9, 0.95\}$ |
| second momentum | $\beta_2$ | $\{0.999\}$ |
| epsilon | $\epsilon$ | $\{1e-8\}$ |

We did not vary the first or second momentum much, since [30] states that the values chosen are already good defaults.

*SGD*:

| hyperparameter | symbol | values |
|---|---|---|
| learning rate | $\lambda$ | $\{0.1, 0.01, 0.001, 0.0001\}$ |
| momentum | $\alpha$ | $\{0.85, 0.9, 0.95\}$ |

*RMSProp*:

| hyperparameter | symbol | values |
|---|---|---|
| learning rate | $\lambda$ | $\{0.1, 0.01, 0.001, 0.0001\}$ |
| discounting factor | $f$ | $\{0.9, 0.95\}$ |
| epsilon | $\epsilon$ | $\{1e-8\}$ |

*PAL*:

| hyperparameter | symbol | values |
|---|---|---|
| measuring step size | $\mu$ | $\{10^0, 10^{-0.5}, 10^{-0.1}, 10^{-0.15}\}$ |
| direction adaptation factor | $\beta$ | $\{0, 0.4\}$ |
| update step adaptation | $\alpha$ | $\{1, \frac{1}{0.8}\}$ |
| maximum step size | $s_{max}$ | $\{10^{0.5}(\approx 3.16)\}$ |

In our implementation we worked with a inverse update step adaptation $\gamma = \frac{1}{\alpha}$.

*SLS*:

| hyperparameter | symbol | values |
|---|---|---|
| initial step size | $\mu$ | $\{0.1, 1\}$ |
| step size decay | $\beta$ | $\{0.9, 0.99\}$ |
| step size reset | $\gamma$ | $\{2.0, 2.5\}$ |
| Armijo constant | $c$ | $\{0.1, 0.01\}$ |
| maximum step size | $\mu_{max}$ | $\{10.0\}$ |

*ALIG*:

| hyperparameter | symbol | values |
|---|---|---|
| maximal learning rate | $\lambda$ | $\{10, 1.0, 0.1, 0.01\}$ |
| momentum | $\beta$ | $\{0.85, 0.9, 0.95\}$ |

*COCOB*:

| hyperparameter | symbol | values |
|---|---|---|
| restriction factor | $\alpha$ | $\{25, 50, 75, 100, 125, 150, 175, 200\}$ |

*SGDHD*:

| hyperparameter | symbol | values |
|---|---|---|
| learning rate | $\lambda$ | $\{0.1, 0.01, 0.001\}$ |
| hyper gradient learning rate | $\beta$ | $\{0.1, 0.01, 0.001, 0.0001\}$ |

## D.9 Detailed numerical results

Table 2: Performance comparison of *PAL*, *RMSProp*, *ADAM*, *COCOB*, *SGDHD*, *ALIG* and *SGD*. All hyperparameter combinations given in Appendix D.8 were evaluated for each architecture. Results are averaged over 3 runs starting from different random seeds, except for training on ImageNet, for which results were not averaged. Note that tests on Imagenet were performed with the best hyperparameters found on CIFAR-100 to test the transferability of hyperparameters. Medians an Quartiles describe the distribution of results over reasonable hyper-parameter ranges.

| dataset | network | optimizer | training loss min | median; p25; p75 | validation accuracy max | median; p25; p75 | test accuracy max | median; p25; p75 |
|---|---|---|---|---|---|---|---|---|
| CIFAR-10 | EfficientNet | COCOB | 0.659 | 0.824; 0.739; 0.855 | 0.857 | 0.837; 0.832; 0.845 | 0.843 | 0.824; 0.818; 0.832 |
| | | ALIG | 0.279 | 0.89; 0.464; 1.911 | 0.906 | 0.805; 0.451; 0.895 | 0.893 | 0.757; 0.297; 0.878 |
| | | SGDHD | 2.002 | 6.239; 4.357; 7.803 | 0.834 | 0.657; 0.18; 0.74 | 0.828 | 0.647; 0.179; 0.731 |
| | | SLS | 2.837 | 5.596; 4.681; 6.292 | 0.653 | 0.357; 0.211; 0.443 | 0.643 | 0.357; 0.216; 0.442 |
| | | RMSP | 0.154 | 0.637; 0.333; 1.261 | **0.93** | 0.864; 0.658; 0.902 | 0.919 | 0.854; 0.648; 0.889 |
| | | ADAM | 0.155 | 0.818; 0.292; 2.275 | 0.926 | 0.841; 0.211; 0.907 | 0.919 | 0.83; 0.1; 0.896 |
| | | SGD | 0.165 | 2.287; 0.343; 4.221 | **0.93** | 0.872; 0.794; 0.915 | **0.921** | 0.862; 0.784; 0.906 |
| | | PAL | **0.137** | **0.244**; 0.186; 0.388 | 0.927 | **0.912**; 0.906; 0.921 | 0.916 | **0.902**; 0.889; 0.908 |
| CIFAR-10 | MobileNetV2 | COCOB | 0.232 | **0.282**; 0.257; 0.295 | 0.879 | 0.87; 0.866; 0.876 | 0.865 | 0.852; 0.848; 0.865 |
| | | ALIG | 0.183 | 0.938; 0.347; 1.926 | 0.914 | 0.695; 0.233; 0.888 | 0.897 | 0.528; 0.1; 0.851 |
| | | SGDHD | 0.698 | 2.234; 1.835; 4.366 | 0.886 | 0.75; 0.298; 0.807 | 0.877 | 0.737; 0.295; 0.791 |
| | | SLS | 1.387 | 2.462; 2.011; 2.584 | 0.667 | 0.443; 0.407; 0.504 | 0.595 | 0.4; 0.343; 0.437 |
| | | RMSP | **0.085** | 0.493; 0.337; 0.918 | 0.938 | 0.872; 0.675; 0.895 | 0.929 | 0.865; 0.664; 0.882 |
| | | ADAM | 0.095 | 0.477; 0.314; 1.861 | 0.939 | 0.874; 0.309; 0.896 | 0.93 | 0.864; 0.289; 0.886 |
| | | SGD | 0.149 | 0.878; 0.204; 1.552 | **0.947** | **0.907**; 0.87; 0.933 | **0.94** | **0.899**; 0.859; 0.925 |
| | | PAL | 0.15 | 0.377; 0.205; 0.531 | 0.92 | 0.905; 0.896; 0.91 | 0.905 | 0.886; 0.877; 0.896 |
| CIFAR-10 | DenseNet40 | COCOB | 0.228 | 0.234; 0.23; 0.24 | 0.907 | 0.903; 0.901; 0.904 | 0.894 | 0.889; 0.885; 0.892 |
| | | ALIG | 0.188 | 0.604; 0.227; 2.903 | 0.918 | 0.848; 0.438; 0.902 | 0.902 | 0.784; 0.336; 0.875 |
| | | SGDHD | 1.094 | 2.279; 1.349; 2.908 | 0.775 | 0.341; 0.099; 0.696 | 0.762 | 0.1; 0.1; 0.26 |
| | | SLS | **0.065** | **0.115**; 0.104; 0.189 | 0.91 | 0.904; 0.897; 0.905 | 0.901 | **0.893**; 0.89; 0.897 |
| | | RMSP | 0.147 | 0.398; 0.256; 0.915 | 0.927 | 0.879; 0.737; 0.915 | 0.92 | 0.867; 0.717; 0.909 |
| | | ADAM | 0.138 | 0.749; 0.274; 1.028 | 0.922 | 0.777; 0.611; 0.91 | 0.913 | 0.806; 0.605; 0.907 |
| | | SGD | 0.147 | 0.794; 0.396; 1.746 | **0.932** | 0.855; 0.537; 0.914 | **0.93** | 0.847; 0.528; 0.91 |
| | | PAL | 0.099 | 0.217; 0.165; 0.343 | 0.925 | **0.907**; 0.894; 0.919 | 0.916 | 0.882; 0.861; 0.9 |
| CIFAR-10 | ResNet32 | COCOB | 0.125 | 0.128; 0.127; 0.129 | 0.888 | 0.886; 0.885; 0.887 | 0.878 | 0.872; 0.871; 0.874 |
| | | ALIG | 0.122 | 0.658; 0.279; 1.485 | 0.892 | 0.815; 0.47; 0.881 | 0.866 | 0.71; 0.367; 0.852 |
| | | SGDHD | 0.35 | 0.464; 0.413; 0.701 | 0.864 | 0.835; 0.791; 0.843 | 0.837 | 0.796; 0.766; 0.827 |
| | | SLS | **0.005** | **0.006**; 0.005; 0.827 | 0.871 | 0.856; 0.758; 0.869 | 0.846 | 0.824; 0.657; 0.839 |
| | | RMSP | 0.105 | 0.199; 0.129; 0.498 | 0.922 | 0.884; 0.804; 0.904 | 0.915 | 0.877; 0.792; 0.896 |
| | | ADAM | 0.105 | 0.332; 0.133; 1.004 | 0.917 | 0.875; 0.677; 0.881 | 0.914 | 0.868; 0.654; 0.873 |
| | | SGD | 0.098 | 0.131; 0.118; 0.322 | **0.939** | **0.899**; 0.85; 0.924 | **0.933** | **0.893**; 0.838; 0.92 |
| | | PAL | 0.05 | 0.105; 0.075; 0.195 | 0.921 | 0.893; 0.887; 0.906 | 0.903 | 0.88; 0.849; 0.888 |
| CIFAR-100 | DenseNet40 | COCOB | 0.739 | 0.761; 0.75; 0.772 | 0.642 | 0.633; 0.631; 0.637 | 0.646 | 0.632; 0.629; 0.637 |
| | | ALIG | 0.488 | 2.125; 0.988; 3.128 | 0.637 | 0.508; 0.391; 0.623 | 0.616 | 0.48; 0.264; 0.605 |
| | | SGDHD | 1.78 | 2.6; 2.179; 3.465 | 0.566 | 0.418; 0.274; 0.504 | 0.55 | 0.296; 0.159; 0.497 |
| | | SLS | 1.367 | 1.908; 1.446; 1.96 | 0.719 | 0.593; 0.572; 0.698 | 0.612 | 0.479; 0.422; 0.554 |
| | | RMSP | 0.348 | 1.238; 0.78; 1.972 | 0.716 | 0.583; 0.481; 0.634 | 0.712 | 0.588; 0.482; 0.631 |
| | | ADAM | 0.326 | 1.114; 0.859; 3.53 | 0.715 | 0.601; 0.165; 0.637 | 0.712 | 0.599; 0.226; 0.641 |
| | | SGD | 0.376 | 0.713; 0.431; 2.154 | 0.75 | 0.633; 0.489; 0.709 | **0.753** | 0.634; 0.498; 0.708 |
| | | PAL | **0.275** | **0.376**; 0.312; 0.459 | 0.73 | **0.686**; 0.66; 0.705 | 0.717 | **0.676**; 0.642; 0.695 |
| CIFAR-100 | EfficientNet | COCOB | 0.802 | 0.817; 0.807; 0.822 | 0.594 | 0.583; 0.581; 0.59 | 0.596 | 0.582; 0.58; 0.588 |
| | | ALIG | 0.57 | 2.4; 0.995; 4.085 | 0.612 | 0.494; 0.169; 0.6 | 0.599 | 0.458; 0.115; 0.597 |
| | | SGDHD | 3.545 | 6.528; 5.519; 8.917 | 0.529 | 0.337; 0.178; 0.463 | 0.513 | 0.342; 0.179; 0.468 |
| | | SLS | 3.731 | 6.713; 6.348; 6.857 | 0.474 | 0.212; 0.208; 0.227 | 0.375 | 0.203; 0.149; 0.208 |
| | | RMSP | 0.422 | 1.823; 1.253; 2.968 | 0.675 | 0.517; 0.383; 0.588 | 0.678 | 0.521; 0.382; 0.59 |
| | | ADAM | 0.45 | 1.394; 1.312; 4.606 | 0.684 | 0.518; 0.025; 0.619 | 0.684 | 0.524; 0.01; 0.621 |
| | | SGD | 0.42 | 2.44; 0.633; 5.214 | **0.712** | 0.579; 0.473; 0.661 | **0.709** | 0.579; 0.476; 0.658 |
| | | PAL | **0.372** | **0.471**; 0.409; 0.772 | 0.693 | **0.666**; 0.638; 0.676 | 0.69 | **0.664**; 0.63; 0.671 |
| CIFAR-100 | MobileNetV2 | COCOB | 0.486 | **0.513**; 0.492; 0.536 | 0.644 | 0.63; 0.626; 0.638 | 0.644 | 0.63; 0.623; 0.638 |
| | | ALIG | 0.323 | 2.396; 0.817; 4.247 | 0.661 | 0.41; 0.034; 0.623 | 0.652 | 0.229; 0.01; 0.602 |
| | | SGDHD | 1.485 | 3.307; 2.425; 7.002 | 0.593 | 0.476; 0.39; 0.545 | 0.589 | 0.456; 0.385; 0.525 |
| | | SLS | 3.857 | 5.086; 5.031; 5.64 | 0.332 | 0.2; 0.099; 0.203 | 0.197 | 0.081; 0.052; 0.126 |
| | | RMSP | 0.198 | 1.518; 0.718; 3.368 | 0.728 | 0.593; 0.43; 0.635 | 0.727 | 0.593; 0.431; 0.634 |
| | | ADAM | 0.218 | 1.873; 0.776; 4.524 | 0.729 | 0.528; 0.025; 0.593 | 0.729 | 0.533; 0.02; 0.595 |
| | | SGD | 0.4 | 0.974; 0.473; 2.151 | **0.733** | 0.657; 0.57; 0.7 | **0.736** | 0.659; 0.573; 0.701 |
| | | PAL | **0.181** | 0.602; 0.314; 1.571 | 0.726 | **0.666**; 0.574; 0.689 | 0.722 | **0.664**; 0.509; 0.681 |
| CIFAR-100 | ResNet32 | COCOB | 0.498 | 0.569; 0.524; 0.673 | 0.609 | 0.608; 0.607; 0.608 | 0.605 | 0.602; 0.599; 0.604 |
| | | ALIG | 0.537 | 1.932; 0.995; 3.572 | 0.597 | 0.491; 0.19; 0.58 | 0.587 | 0.414; 0.144; 0.549 |
| | | SGDHD | 0.881 | 1.359; 1.06; 1.772 | 0.601 | 0.539; 0.472; 0.586 | 0.599 | 0.517; 0.431; 0.571 |
| | | SLS | 2.62 | 2.808; 2.78; 2.82 | 0.399 | 0.388; 0.384; 0.392 | 0.363 | 0.305; 0.274; 0.33 |
| | | RMSP | 0.519 | 1.019; 0.807; 2.083 | 0.661 | 0.599; 0.455; 0.651 | 0.656 | 0.603; 0.455; 0.65 |
| | | ADAM | 0.402 | 1.772; 0.768; 3.038 | 0.659 | 0.513; 0.262; 0.564 | 0.658 | 0.519; 0.255; 0.567 |
| | | SGD | 0.375 | **0.474**; 0.4; 1.522 | **0.697** | 0.614; 0.494; 0.672 | **0.694** | 0.616; 0.502; 0.667 |
| | | PAL | **0.339** | 0.485; 0.369; 1.424 | 0.662 | **0.636**; 0.546; 0.652 | 0.663 | **0.621**; 0.512; 0.647 |

| Dataset | Model | Optimizer | | | | | | |
| --- | --- | --- | --- | --- | --- | --- | --- | --- |
| TOLSTOI | RNN | COCOB | 1.506 | 1.56; 1.533; 1.593 | 0.589 | 0.58; 0.573; 0.584 | 0.582 | 0.572; 0.566; 0.577 |
| | | ALIG | 1.501 | 1.562; 1.528; 1.766 | 0.591 | 0.579; 0.523; 0.586 | 0.584 | 0.571; 0.513; 0.577 |
| | | SGDHD | 2.282 | 2.433; 2.379; 2.445 | 0.375 | 0.338; 0.336; 0.348 | 0.369 | 0.334; 0.332; 0.344 |
| | | SLS | 3.128 | 3.149; 3.136; 3.156 | 0.169 | 0.159; 0.158; 0.165 | 0.168 | 0.158; 0.157; 0.164 |
| | | RMSP | **1.475** | **1.509**; 1.492; 1.556 | **0.599** | **0.591**; 0.579; 0.595 | **0.592** | **0.583**; 0.572; 0.587 |
| | | ADAM | 1.516 | 1.655; 1.596; 1.681 | 0.588 | 0.567; 0.55; 0.578 | 0.581 | 0.561; 0.543; 0.571 |
| | | SGD | 1.496 | 1.872; 1.56; 2.675 | 0.594 | 0.483; 0.278; 0.573 | 0.587 | 0.476; 0.275; 0.566 |
| | | PAL | 1.528 | 1.569; 1.547; 1.588 | 0.587 | 0.581; 0.577; 0.586 | 0.579 | 0.571; 0.556; 0.575 |
| Imagenet | ResNet50 | RMSP | 9.485 | – | 0.286 | – | 0.28 | – |
| | | ADAM | 1.863 | – | 0.562 | – | 0.559 | – |
| | | SLS | 3.808 | – | 0.286 | – | 0.069 | – |
| | | SGD | 1.123 | – | **0.656** | – | **0.65** | – |
| | | PAL | **0.773** | – | 0.608 | – | 0.608 | – |
| Imagenet | DenseNet121 | RMSP | 6.901 | – | 0.0 | – | 0.0 | – |
| | | ADAM | 6.901 | – | 0.001 | – | 0.0 | – |
| | | SLS | 7.768 | – | 0.001 | – | 0.001 | – |
| | | SGD | 3.308 | – | 0.458 | – | 0.452 | – |
| | | PAL | **1.228** | – | **0.617** | – | **0.611** | – |

# E   Binary Line Search

The optimal binary line search we compared *PAL* against. Since the line decreases in negative gradient direction, at first a extrapolation phase performs as many steps forward as the loss does not increase. Afterwards a binary search is performed. This approach is valid if the underlying line is convex. For simple readability we chose Python 3.6 syntax.

```
Input: max_num_of_search_steps                                               1
def binary_line_search(last_loss, step, counter, is_extrapolate):            2
    if counter == max_num_of_search_steps:                                   3
        return last_loss                                                     4
    counter += 1                                                             5
    if is_extrapolate:                                                       6
        current_loss = do_step_on_line(step)                                 7
        if current_loss < last_loss:                                         8
            return binary_line_search(current_loss, step,counter,            9
                is_extrapolate)
        else:                                                               10
            is_extrapolate = False                                          11
            do_step_on_line(-step,get_loss=False)                           12
    if not is_extrapolate:                                                  13
        loss_right = do_step_on_line(0.5*step, True)                        14
        if loss_right < last_loss:                                          15
            return binary_line_search(loss_right,                           16
                0.5*step,counter, is_extrapolate)
        loss_left = do_step_on_line(-1*step, True)                          17
        if loss_left < last_loss:                                           18
            return binary_line_search(loss_left,                            19
                0.5*step,counter, is_extrapolate)
        do_step_on_line(0.5*step,get_loss=False)                            20
        if loss_right >= last_loss and loss_left >= last_loss:              21
            return binary_line_search(loss_left, 0.5*step,                  22
                counter, is_extrapolate)
        else:                                                               23
            # this state is not possible                                    24
```