[Reviews · NeurIPS 2020]

Review 1

Summary and Contributions: This paper proposes a line search method for training deep networks. They propose locally approximating the loss surface with a parabola using two estimates of the loss on a mini-batch and then inferring the step size based on this approximation. They provide evidence that this approximation is reliable on a number of datasets and for a number of models. They also provide evidence that their approximate line search is in fact performing better than exact line search as well as previous line search methods and is better or competitive with SGD in converged training and validation loss/accuracy. Their method is however not beating the generalization performance of SGD on some datasets and is not as approximately twice as slow as SGD. Some suggestions are given below.

Strengths: - Fig1 is interesting and supports the claim of the paper on parabolic estimate of the loss. What happens closer to the end of the training? After 50K iterations on cifar-10 and two learning rate drops? In appendix EfficientNet and MobileNet seem not to always follow the assumption (figure 8 and 9 top left subfigures). How often does that happen? - Fig 4: The learning rate curves for the proposed method seems fairly smooth. Have you tried reusing the learning rate from previous steps to reduce the computational cost? Or is the smoothness an artifact of the averaging over 3 runs? - Fig 5-6: The curves supporting the claim that exact line search is not desirable are fairly convincing. It would be nice to do a hyperparameter study of the proposed method where the estimated step size is scaled by a constant and plot the performance as a function of this hyperparameter. Do we need to always underestimate or do we need to sometimes overestimate the step size?

Weaknesses: - Fig 4: The proposed method does not generalize as well as SGD. Have you used weight decay? - Fig 4: The generalization gap between Adam and SGD should mostly go away with appropriate weight decay. See the following paper: Loshchilov and Hutter, Decoupled Weight Decay Regularization, ICLR 2019 Zhang et al, Three Mechanisms of Weight Decay Regularization, ICLR 2019 - Fig 4: the comparison with SGD on imagenet does not seem fair as we know there exists a learning rate schedule that works well. Also it’s not clear why other methods are not shown on imagenet.

Correctness: I have not found any errors. I have not checked the theoretical analysis.

Clarity: - Fig 2 Can you also show similar plots for the last epoch? What are the bars? Box plot is supposed to show min/max/median/Q1/Q3 but the min-max are not covering the min-max of drawn points. - Line 38: The definition of x seems to be a confusing notation abuse. Please consider a different notation for samples vs mini-batches. - Line 44: well enough?

Relation to Prior Work: The related works are mentioned and compared to. It would be nice to discuss briefly the algorithm of previous methods and more details in the method and experiments section.

Reproducibility: Yes

Additional Feedback: Typos: 227 (contrasts with), 33 (fits with), 51 (contrasts?), 59 (beautiful?!), 82 (commonly) ======== After rebuttal: I thank the authors for their detailed response. I do not see any major problems with the current results of the paper and additional plots provided in the rebuttal. I believe empirical results without theoretical justification have merits. The suggestions by other reviewers for more empirical evidence for example with additional loss functions is nice to have but in my opinion does not undermine current results. As such, I am increasing my score to 7. I encourage the authors to incorporate their rebuttal response into the paper.


Review 2

Summary and Contributions: After considering the authors, response (rebuttal) I have a slightly more positive view of the paper, but the increase was not enough to change my overall score of the paper. This paper presents a parabolic-approximation line search approach to chose step-sizes and that is claimed to be suitable for deep learning optimization. The major contributions include: Contribution 1: Empirical analysis suggesting convexity of the loss function in the negative gradient direction and that parabolic approximations of the empirical loss function are well suited to estimate the minima in these directions. (significance: medium) Contribution 2: A-line search procedure based on the parabolic fit to the loss function at one point in addition to the current point along the current step direction that can be used to compute a "good" step-size. (significance: medium) Contribution 3: Empirical comparison of the proposed method with known step-size schedules for first-order methods including SGD, Adam, RMSP, and a stochastic line search method proposed in [54]. (significance: medium) Contribution 4: Convergence analysis under very strong assumptions. (significance: very low)

Strengths: - Significance and novelty of the contributions: A ) Contribution 1: Empirical evidence of the observed convexity property is presented for classification tasks on CIFAR-10 using ResNet32, DenseNet40, MobileNetV2, and EfficientNet. The convexity property aspect was also observed in the Probabilistic Line Search paper [36] on MNIST but was not been extensively studied empirically in that work. B ) Contribution 2: The novelty of the proposed line search procedure is that it requires computing only one additional sample loss measurement (in addition to the loss function value and, the gradient at the current point) to fit parabolic approximation and estimate the step-size. We note that using a parabolic fit is a standard approach in line search procedures in deterministic optimization. B ) Contribution 3: The method is compared to SGD, Adam, RMSP with known "good" schedules and SLS method on CIFAR-10, CIFAR-100, and ImageNet classification tasks using various architectures. The paper documents the ability of the method to pick good learning rates, decreasing the training loss faster (iteration-wise) than SGD, Adam, and RMSP in some settings. However, it never outperforms SGD in terms of val-accuracy. An RNN task on TOLSTOI is also presented in the appendix. B ) Contribution 4: The analysis seems sound to me, although it requires very strong assumptions. - Relevance to the NeurIPS community: Clearly related to the NeurIPS community. Choosing step-size schedules is vital for deep learning and allows for faster tuning of models. This work aims at providing a method to automatically select step-sizes.

Weaknesses: - Weaknesses and limitations of the contributions: A ) Contribution 1: Empirical evidence of the observed convex parabolic shape is only presented for classification tasks on one dataset CIFAR-10. Since this is the main contribution of the paper, other deep learning tasks and loss functions should be explored. Other datasets should also be studied to support the claim. (The author's feedback has adequately addressed this issue.) B ) Contribution 2: The method introduces new hyperparameters (update step adaptation, measuring step size, maximal step size). The sensitivity study in Figure 14 is not convincing for the following reasons: - it's unclear to me how different combinations of these parameters would perform based on that figure. - Although the gradient is normalized, I suspect the measuring step size value will still depend on the scale of the problem and I am concerned that the proposed range in Figure 14 is only adapted to ResNet32 trained on the CIFAR-10 problem. The method is claimed to be generalizable to any step direction, but no empirical evidence is presented to back up this up. It would be interesting to see how the proposed step size procedure would perform on SGD, with or without momentum and Adam directions. (The author's feedback has addressed this issue.) B ) Contribution 3: The authors used a so-called conjugate gradient method to pick the search direction, making it difficult to access whether PAL picks good learning rates based on the figures. It would be more convincing to compare the learning rates obtained by PAL using SGD-with-momentum (or any other algorithm) directions to the optimal learning rate schedule for the same algorithm. - Validation accuracy is consistently lower than SGD across the presented problems. - No plots comparing CPU times are included in the experiments. Computing an additional evaluation of the loss function on every step requires an additional forward pass and how this effects the total run time should be presented. - Comparison with PLS is not included. (The author's feedback has adequately addressed this issue.) - Comparison with second-order optimizers is not included. B ) Contribution 4: The convergence proof is provided under strong assumptions (parabolic shape + same Q matrix for individual-loss) which are, as mentioned by the authors, not valid for general deep learning scenarios.

Correctness: Claims and method seem correct to me for the presented settings in the paper. Empirical methodology seems correct, but can definitely be improved and extended as described in the Weakness part.

Clarity: Yes

Relation to Prior Work: Differences with previous line search procedures are clearly discussed. However, local curvature based methods such as Barzilai-Borwein step size [1] Stochastic Polyak step size [2] and Nestrov adaptive step size [3] to select the step sizes are not mentioned in the discussion of related work. [1] Liang, Jinxiu & Xu, Yong & Bao, Chenglong & Quan, Yuhui & Ji, Hui. (2019). Barzilai-Borwein-based Adaptive Learning Rate for Deep Learning. Pattern Recognition Letters. 128. 10.1016/j.patrec.2019.08.029. [2] Loizou, Nicolas & Vaswani, Sharan & Laradji, Issam & Lacoste-Julien, Simon. (2020). Stochastic Polyak Step-size for SGD: An Adaptive Learning Rate for Fast Convergence. [3] Bahamou, Achraf & Goldfarb, Donald. (2019). A Dynamic Sampling Adaptive-SGD Method for Machine Learning.

Reproducibility: Yes

Additional Feedback: The paper refers to the step direction used in the PAL algorithm as a conjugate (gradient) direction. It would be more correct to refer to it as an conjugate (gradient)-"like" direction, since it adds a fixed multiple of the previous direction to the (stochastic) negative gradient rather than computing the appropriate multiple using Fletcher-Reeves, etc. Additional feedback after reviewing Authors' feedback: The paper never specifies the loss function used in the empirical experiments. I assume that for all problems it was softmax. Consequently, I believe that the empirical justification of the "parabolic" step determination approach only applies to problems with softmax loss functions that are solved on various CNNs. There is no empirical support that the "parabolic" step length approach would work well on other loss functions or other types of DNNs. I still have a negative view of the theoretical contribution of the paper, and believe that the paper would be improved if the theoretical analysis section were removed from the main body of the paper, and only mentioned briefly in the appendix, while indicating that rather strong assumptions were being made to obtain the propositions.. My view of the soundness of the paper's extensive empirical work is now somewhat higher and I seriously considered raising my score for the paper. However, in the end I felt that my original score of 5 was still the appropriate one to assign the paper..


Review 3

Summary and Contributions: This paper presents an interesting empirical analysis on the loss landscape of commonly-used deep learning loss functions. It introduces a novel algorithm called PAL that automatically sets the step size. PAL’s step size is obtained from a univariate quadratic approximation of the loss along the (subsampled) negative gradient direction. The choice of quadratic approximation is based on the empirical observation that locally, the subsampled loss along the subsampled negative gradient direction is often convex. In most experiments presented, PAL performs better than popular stochastic first-order optimizers, including the state-of-the-art stochastic line search algorithm. The authors provide some theoretical justification to PAL, although the applicability of the theory is very limited. Finally, an investigation of exact line searches on the subsampled losses is conducted, which empirically shows that PAL can often find a local minimum of the univariate function, so descent is often guaranteed on the subsampled loss.

Strengths: The paper is strong in the following aspects: 1. The proposed algorithm PAL performs quite well in terms of convergence rate and generalization properties on multiple deep learning benchmarks. 2. Although PAL requires two forward passes and one backward pass at every iteration to compute its step size, it is nonetheless a fixed cost as it does not require backtracking, which may require more than two forward passes. This makes PAL easier to implement and can be cheaper in terms of computation, compared to line search methods that rely on backtracking. 3. The authors point out (in line 128) that when multiple forward passes are required, model evaluation with random components such as dropout needs to re-use the random state to ensure the same batch loss function is being computed. This is a valuable reminder for readers who wish to use these algorithms.

Weaknesses: The main weaknesses of this paper lie in the soundness of the claims and the clarity of the presentation (see Clarity section for the latter). The major issues are: 1. In section 4.3, the authors introduce “conjugate directions” (Eq.3) to their update, which is essentially just the SGD update with a heavy-ball momentum term. The connection to conjugate gradient is also unjustified. Two directions u and v are said to be conjugate with respect to some positive definite matrix A if <u, Av> = 0. The authors claim that d_t defined in Eq.3 is a conjugate direction, but did not show this for the successive updates, if it is even possible. 2. Related to the above, it is also not clear why the momentum term is needed. This work presents an empirical analysis of the convex loss landscape in the negative (subsampled) gradient direction; however, the loss landscape in the direction with momentum does not seem to be explored. 3. It is also unclear whether this momentum term could be a confounding factor in the comparison between PAL and SLS, as the vanilla version of SLS is just stochastic line search applied to SGD without momentum. 4. In line 107, the authors assume l_t (defined in Eq. 1) is a quadratic function. First of all, this is a strong assumption as it does not hold globally unless the loss functions L are chosen to be the squared loss. For nonconvex functions, it may be true locally, in which case lines 107-108 should use a different notation for the univariate quadratic approximation, rather than overloading l_t. The next few points are regarding the theoretical analysis presented in section 4.4. 5. Assumption 2 is way too strong for all step sizes and an arbitrary loss function, especially when the focus of this paper is deep learning objectives. It is also only justified locally throughout the paper by plotting the univariate function on sampled iterates. 6. Lemma 1, which says if every slice of a multivariate function is univariate quadratic, then the overall function is also quadratic. This is rather trivial, and it would be much cleaner to simply say that the theoretical analysis is based on the squared loss. 7. Proposition 2 assumes that the components in the finite-sum objective are all quadratic functions, and that the positive definite matrices defining them are all the same. Not only is it trivial that in this case, the minimizer of the overall objective and the minimizers of the individual objectives coincide, it also implies that for a typical machine learning application all the features are the same, but the labels may be different.

Correctness: The empirical methodology is mostly correct. The specific errors and limitations of the arguments are listed in the Weaknesses and Clarity sections.

Clarity: The mathematical formulation contains multiple errors and inconsistencies, which can be improved for a clearer presentation. Specifically: 1. In line 38, the objective function is parameterized in terms of \theta, the dimensionality of which is usually not the same as the number of batches n, therefore L should not be a function mapping from R^n to R. 2. Related to the above point: in proposition 2, the number of batches is denoted by m, instead of n, which is used to denote the number of parameters in Assumption 2, contradicting the original setup. It would be better to keep the notation consistent. 3. Typo: the \theta in line 40-41 should also contain a subscript t. 4. The univariate function l_t defined in Eq.1 should not be referred to as a line, even for simplicity. It is just some nonconvex function defined on the batch loss along the update direction, parameterized by the possible step sizes. Referring it to a line can be confusing as it’s more natural to associate it with the line given by the span of the update direction. In particular, the term “line” used Assumption 1 is actually referring to the update direction instead of the univariate function, thus overloading the term. 5. When deriving the PAL line-search rule in section 4.1, it may be clearer to begin by saying that ``the step size is computed by minimizing a second-order Taylor approximation around l_t(0)``. This will allow the authors to justify the choice of a, b, and c in lines 112-113. 6. Typo in line 136: s_{upt} should be s_{upd} 7. In line 229, the authors claim that the experiments from SLS have been reproduced on ResNet34, referencing Appendix D.1. Should this be Appendix D.3 instead?

Relation to Prior Work: The comparison to previous works is thorough and well-written, although it would better if the authors could also address the following points: 1. To the best of my knowledge, [54] does not make any assumptions on how well the step size found using the stochastic Armijo condition approximates what would’ve been obtained on the deterministic Armijo condition. Therefore the statement on line 56-57 is incorrect. 2. In line 65-66, the authors mention that [9] also analyzes different line search approximations as this work; it would be better if elaboration of the novelties compared to [9] is also provided.

Reproducibility: Yes

Additional Feedback: ======== Update after author feedback ======= As the authors have pointed out, I should not be focusing my evaluation on the theoretical results. However, I still believe the theoretical results are insignificant in terms of contribution. On the empirical side: I had a concern about why the momentum/"conjugate gradient modification" was needed, as the empirical analysis of the convex landscape was performed using the (subsampled) gradient direction. The authors claim in the feedback that the parabolic property is still valid in the momentum/"CG" directions. However, without providing evidence I am not convinced that this claim is generally true. Regarding my concern about whether this momentum/"CG" term \beta could be a confounding factor in their comparison against SLS, the authors claim that this has minor influence. They mentioned in the rebuttal that in most of their best-performing results, \beta's grid-search values are in fact 0 (assuming I'm interpreting "achieved without it" correctly). It would be nice to list out in which cases was it actually 0 for a better comparison against SLS (without momentum). Based on the empirical merits of this paper showing mostly parabolic landscapes for training DNNs, I am happy to increase my score to 5.

[Author Response · NeurIPS 2020]

1 **First of all I would like to thank all reviewers for their valuable and unexpectedly detailed reviews!**
2 **R1,R2,R4:** Figure 1 shows in addition to Figure 2 of the paper that the **parabolic property is also valid on**
3 **further datasets and at the end of the training**. It fits best for MNIST and worsens for more complex tasks.

Figure 1: Direction angles at update step position over training steps.

**R1:** I used a common **weight decay** factor $10^{-4}$ for all experiments as described in appendix D.5.3. I considered searching for an optimal weight decay factor, but omitted it, since 1494 networks already had to be trained for the evaluation. The **smoothness of the learning rate curves**

originates from averaging over epochs which is a standard procedure for plots against epochs. Figure 2 exemplarily shows that on a step wise scale the variance of the learning rate is larger. EfficientNet and MobileNet **contradict the parabolic assumption** only for about the first 500 steps. A **Hyperparameter study** is given in Appendix D.4. My **Box plots** are standard, the circles show outlier outside the interquartile range.

Figure 2: Comparison of PAL to Probabilistic Line Search [36].

**R2:** There is no easy to use implementation for **PLS [36]**. The sum of squared gradients has to be derived manually for each layer, which is a considerable amount of work for modern architectures but after 2 weeks o work I managed to do it for

a ResNet32 see (Figure 2). Comparisons with **second-order methods** are not applicable since the 11GB memory of my graphic cards are not enough to save the Hessian or one of its approximations for the networks investigated.

A comparison of **CPU time** is given in Appendix D.2. The "conjugate" gradient factor has minor influence and most of the best results are achieved without it. Therefore, pure **SGD using PAL's schedule** would mostly exactly train like PAL. In general, however, the optimal schedule depends on the path an optimizer takes on the loss landscape, thus

Figure 3: Sensitivity of the measuring step size $\mu$ of PAL for several Models on CIFAR-10.

**an optimal schedule** has to be directly inferred during the training process and should generally **not be transferable**. Figure 3 shows that $\mu$ **has also a low sensitivity for other networks**.

**R1,R2,R4**: Although the best possible **validation accuracy** to be achieved is an important property, I want to stress that in practice the **robustness** of the hyperparameters is even more important. Thus, please focus your attention on the Box plots on the right of Figure 4,10,11,12. These show that for commonly used hyperparameters, **PAL has a better median for the validation accuracy than SGD** . This is important for scenarios where one has to train very long and cannot try lots of hyperparameter combinations. Note, that for robustness evaluations one should handle the datasets as if they were unknown and one should perform an evaluation with default hyperparameters that a common researcher would chose. Furthermore, several networks should be considered. **This robustness evaluation is one ot the strengths of my work and rarely seen in such detail**. Out of this perspective using ImageNet to check the transfer-ability of hyperparameters is a valid approach. Note that Adam, RMSP and SLS completely fail in this scenario (**R1**).

**R1,R2,R4**: The **clearly inappropriate name "conjugate" gradient** is a leftover, since I used approaches such as Fletcher Reeves in the beginning of my research to obtain a new direction which fulfills the conjugate condition if one assumes a quadratic. This approach, however, works as well as a fixed $\beta$, but is much more expensive. We will change this name. We can assure that the "parabolic property" is also valid in those "adapted" gradient directions. The "conjugate" gradient direction does have minor influence and most of the best results are achieved without it, thus it is **not a confounding factor** in the comparison between PAL and SLS (**R4**).

**R4 (also interesting for R2)**: We have already **incorporated** your important and valuable suggestions about the **clarity of our formulations**, which was of minor effort. However, we have a **conflict of viewpoints and interests** here. This work is one of the rather **rare empirical works in optimization** for deep learning which tries to measure information from **real-world loss functions** to exploit these for optimization instead of starting form **theoretical assumptions, that are never fully valid** in practice (e.g. convexity, over-parameterization, lipschitz continuous gradient). **Thus, I consider my approach as equally valuable as the theoretical approaches** and therefore see no problem in having a weak theoretical part and a strong empirical part instead. **Naturally, the empirical findings of papers with a theoretical approach are relatively weak from a practical point of view due to restrictive assumptions**. My weak results for the SLS approach [54] are a perfect example of this. **I have to emphasize**, that if one comes from empirical results, it is **not necessarily given** that one can do an **in-depth theoretical evaluation**. In my case, the parabolic assumption, which is clearly useful empirically, limits the theoretical analysis because it does not satisfy the lipschitz continuous gradient assumption. Hence, we had to use a simpler theoretical model which still provides convergence guarantees and is likely more valid locally than globally. **I want to encourage you to reconsider your grading again since you mostly focused on the "weakest" part of the paper and mostly ignoring my strong empirical contributions.** **R1,R2:I also hope to have provided enough evidence for the other authors to rethink their grading.**

[Meta-Review · NeurIPS 2020]

There was a ton of discussion about this paper between reviewers and area chairs, multiple reviewers improved their view of the paper based on the author response, and I read through the paper in detail myself. I was still conflicted after reading it, but I am leaning towards recommending acceptance. However, I *implore* the authors to carefully consider the issues brought up by R2 and R3 as well as the issues that I bring up below. I believe that every single one of the issues brought up can be fixed, and will be extremely-disappointed if these issues are not addressed in the final version (it would make me regret recommending acceptance, and probably be more-harsh on empirical papers in the future, especially by the same authors). Some specific comments from my read-through of the paper: - I agree with the authors that optimization is a mix of theory and empirical work, and it is completely ok for works to be purely empirical if the experiments are done well. I appreciate that the reviewers were flexible on this issue. - This step-size is well-known in optimization. For example, it's used to propose trial step-sizes as part of the famous line-search of More and Thunte (which I believe is used as a line-search in the standard line-search L-BFGS implementations, and is explicitly discussed in the book of Nocedal and Wright in the line-search chapter. Those works don't suggest to blindly accept the step-size and are using it in the context of a deterministic line-search, but when the method is being discussed there should be prior related work cited. - In my opinion the theoretical results are uninteresting/irrelevant, and including them in the main paper arguably borders on being misleading: -- Under the assumptions of Lemma 1, the algorithm is equivalent to gradient descent with exact line-search on a quadratic function (or CG if you use the standard non-linear CG variants). You could just state this relationship and then cite a source giving convergence in this well-known setting (in fact, we also know convergence *rates* in this setting). -- After Lemma 1 it says that has "no convergence guarantee" with noise. This is mis-leading: if you add noise to Lemma 1 (even with bounded variance) then the given algorithm *does not converge*. Please fix this so we don't need to correct people over the next 'x' number of years like we now correct people regarding convergence of Adam. -- Similarly, even for noise-free objectives I suspect that this method does not converge in general. You really need to emphasize that Lemma 1 is quite a special case because of this equivalence. -- The assumptions in Proposition 2 are possibly the strongest that I have ever seen regarding a finite-sum problem (I would argue that they are even stronger than the "interpolation" assumptions in recent related work), and as pointed by one of the reviewers they make no sense in a machine learning context. I still think it can be worthwhile to include this result, but please add a paragraph saying that these assumptions are unrealistic and would not hold in machine learning applications. You should consider moving this result to the appendix as suggested by one of the reviewers, as naive readers might be mis-led. - I'm ok with including the word "conjugate" in Section 4.3. However, please take the reviewers suggestion that it should be described as conjugate-like. (It would only be conjugate if you had a quadratic and no noise, or you did an exact line-search with no noise.) You may also want consider discussing the Polyak-Ribiere-Plus method and other approaches to guarantee that the non-linear CG methods actually generate a descent direction (I assume you needed to do something like this in the experiments, which is why beta=0 ends up showing up). - I tend to disagree with the reviewer who brings up the issue that the function may not be parabolic when the momentum direction is included. It would be good to add empirical evidence of this, but I see no reason to believe that it wouldn't be locally parabolic in such directions. - As mentioned by one of the reviewers, the paper/experiments needs to tease apart whether the gain is from the step-size or the use the non-linear CG selection of beta. I suspect that both of these help, and that they might work better together, but an ablation study where you test each of these on their own should be included (slightly annoying because you need to choose the step-size to test the non-linear CG method on its own, but I'm sure you can come up with something reasonable). - I was surprised that when SLS was discussed that "over-parameterization" (or "interpolation" or whatever you want to call it) was not discussed at all. SLS was *only* shown to work in this over-parameterized setting, and indeed it *does not* work in under-parameterized settings. This is significant in several ways: -- If SLS is viewed as the main thing to compare to, then it should be stated *whether each model is over-parmaeterized for each dataset*. If there are cases where this is not true, then SLS doesn't seem to be the right method to compare to. -- It might be the case that the *reason that the proposed method works* is also related to over-parameterization. For example, if we added noise to Lemma 1 but assumed over-parameterization then I expect that you could prove that the proposed method does converge (by basically using the arguments in the SLS paper). I think you should either (a) try to prove this result or (b) do experiments on a very-under-parameterized problem (like in the SLS paper) to test this hypothesis regarding why it works at all. -- I would like to see comparisons against other ways to set the step-size that do not rely on the over-parameterization assumption. I am giving the authors the benefit of the doubt here as the empirical work does seem to be carefully done, but in the camera-ready version I expect to see *at least 3* alternate ways to set the step-size among the experiments (some suggestions include coin betting, "no more pesky learning rates", and maybe one of those "do gradient descent on the learning rate" papers like Baydin et al. which often come out but don't work too well when the experiments aren't being run by the author of the paper).